EMBO
Molecular Medicine

# Macrophage deficiency of miR-21 promotes apoptosis, plaque necrosis, and vascular inflammation during atherogenesis

Alberto Canfrán-Duque[1], Noemi Rotllan[1], Xinbo Zhang[1], Marta Fernández-Fuertes[1], Cristina Ramírez-Hidalgo[1], Elisa Araldi[1], Lidia Daimiel[1], Rebeca Busto[2,3], Carlos Fernández-Hernando[1,*] & Yajaira Suárez[1,**]

## Abstract

Atherosclerosis, the major cause of cardiovascular disease, is a chronic inflammatory disease characterized by the accumulation of lipids and inflammatory cells in the artery wall. Aberrant expression of microRNAs has been implicated in the pathophysiological processes underlying the progression of atherosclerosis. Here, we define the contribution of miR-21 in hematopoietic cells during atherogenesis. Interestingly, we found that miR-21 is the most abundant miRNA in macrophages and its absence results in accelerated atherosclerosis, plaque necrosis, and vascular inflammation. miR-21 expression influences foam cell formation, sensitivity to ER-stress-induced apoptosis, and phagocytic clearance capacity. Mechanistically, we discovered that the absence of miR-21 in macrophages increases the expression of the miR-21 target gene, MKK3, promoting the induction of p38-CHOP and JNK signaling. Both pathways enhance macrophage apoptosis and promote the post-translational degradation of ABCG1, a transporter that regulates cholesterol efflux in macrophages. Altogether, these findings reveal a major role for hematopoietic miR-21 in atherogenesis.

**Keywords** apoptosis; atherosclerosis; macrophage polarization; miRNA
**Subject Categories** Immunology; Vascular Biology & Angiogenesis

## Introduction

microRNAs (miRNAs) are an established class of well-conserved, short non-coding RNAs, known to play major roles in most, if not all, biological processes by influencing the stability and translation of mRNAs (Ambros, 2004; Bartel, 2004; Filipowicz *et al*, 2008). The relevance of miRNAs in vascular physiology, lipid metabolism and inflammation, and their implication in vascular and metabolic diseases has been extensively studied (Urbich *et al*, 2008; Araldi *et al*, 2012; Canfran-Duque *et al*, 2014; Rotllan *et al*, 2016). Considerable experimental evidence indicates that cellular and molecular processes related to the development of cardiovascular diseases (CVD), including changes in the function of endothelial cells (ECs), vascular smooth muscle cells (VSMCs), and macrophages/foam cells, are affected by a plethora of miRNAs.

Among these studies, several reports have demonstrated a key role for miR-21 in resolving inflammation and negatively regulating pro-inflammatory responses, including many of the same stimuli that trigger miR-21 induction itself (Sheedy *et al*, 2010; Sheedy, 2015). These include lipopolysaccharide (LPS) and tumor necrosis factor-α (TNF-α), which initiate the inflammatory circuit and prime the cells of the immune system for action. This was linked to the regulation of a proposed negative regulator of interleukin-10 (IL-10) production, programmed cell death 4 (PDCD4), loss of which was shown to protect from LPS-induced lethality (Sheedy *et al*, 2010). Increased miR-21 expression is linked with conditions involving altered immune response including psoriasis, chronic bacterial inflammation, and asthma (Lu *et al*, 2009; Sheedy *et al*, 2010; Guinea-Viniegra *et al*, 2014). These findings suggest that miR-21 dysregulation may contribute to the pathogenesis of these diseases by promoting an anti-inflammatory, immunosuppressive environment. Interestingly, TNF-α has been associated with cell death, and the more recently described "necroptosis" observed in inflammatory macrophages. Therefore, negative regulation of TNF-α levels by miR-21 may dampen excessive inflammation (Sheedy *et al*, 2010; Sheedy, 2015). Notably, reciprocal regulation of TNF-α and miR-21 may be an autoregulatory loop. Indeed, in macrophages, TNF-α can promote miR-21 biogenesis as well as the turnover of PDCD4 (Sheedy *et al*, 2010; Das *et al*, 2014; Sheedy, 2015). Moreover, miR-21 expression has been associated with a "M2-like phenotype",

1   Vascular Biology and Therapeutics Program, Integrative Cell Signaling and Neurobiology of Metabolism Program and the Departments of Comparative Medicine and Pathology, Yale University School of Medicine, New Haven, CT, USA
2   Servicio de Bioquímica-Investigación, Hospital Universitario Ramón y Cajal de Investigación Sanitaria (IRyCIS), Madrid, Spain
3   Centro de Investigación Biomédica en Red Fisiopatología de la Obesidad y Nutrición, Madrid, Spain
*Corresponding author. Tel: +1 203 737 2082; Fax: +1 203 737 2290; E-mail: carlos.fernandez@yale.edu
**Corresponding author. Tel: +1 203 737 8858; Fax: +1 203 737 2290; E-mail: yajaira.suarez@yale.edu

characterized by elevated IL-10 protein levels and increased arginase 1 (Arg1) expression in macrophages (Caescu *et al*, 2015). Interestingly, these alternatively activated macrophages have been associated with atherosclerotic plaque regression in mice, suggesting that pathways that promote this activation have the capability to influence atherogenesis and plaque fate (Feig *et al*, 2011a,b; Moore & Tabas, 2011; Moore *et al*, 2013).

In addition to its role in regulating inflammatory responses, a number of reports have indicated that miR-21 regulates important functions of ECs and VSMCs (Liu *et al*, 2011; Zhou *et al*, 2011; Maegdefessel *et al*, 2012). miR-21 targets peroxisome proliferator-activated receptor-α (PPARα) in a regulatory loop that modulates flow-induced endothelial inflammation. Oscillatory shear stress induces the expression of miR-21 at the transcriptional level in ECs leading to a repression of PPARα expression (Zhou *et al*, 2011). Decreased expression of PPARα reduces the inhibitory effects of PPARα on activator protein 1 (AP-1) activation and, hence, promotes the expression of adhesion molecules and chemoattractant chemokines, including vascular cell adhesion molecule-1 (VCAM-1) and monocyte chemoattractant protein-1 (MCP-1), respectively (Zhou *et al*, 2011). In addition to regulating EC inflammation, a number of reports have shown that miR-21 also regulates angiogenesis, EC survival, and nitric oxide (NO) production (Liu *et al*, 2011). Most of these effects are attributed to the miR-21-mediated downregulation of phosphatase and tensin homolog (PTEN) expression, a phosphatase that negatively regulates protein kinase B (PKB, aka Akt) Akt/PKB signaling pathway. The enhanced activity of Akt observed in ECs overexpressing miR-21 results in augmented phosphorylation and activity of the endothelial nitric oxide synthase (eNOS) and increased EC survival and proliferation (Liu *et al*, 2011). Together, these findings strongly suggest that miR-21 expression might influence the progression of atherosclerosis by regulating EC inflammation, NO production, proliferation, and apoptosis.

The role of miR-21 in vascular remodeling and VSMC functions has been very well established (Ji *et al*, 2007; Davis-Dusenbery *et al*, 2011). Following vascular injury induced by ligation of the carotid artery or balloon angioplasty, as well as in human atherosclerotic lesions, the expression of miR-21 is increased (Ji *et al*, 2007; Cheng & Zhang, 2010). miR-21 expression regulates bone morphogenic proteins (BMPs) via suppression of programmed cell death 4 (PDCD4; Kang *et al*, 2012). Interestingly, transforming growth factor-β (TGFβ) and BMPs increase the expression of miR-21 in VSMCs at the post-transcriptional level by recruitment of mothers against decapentaplegic homolog (SMAD) signal transducers to the microprocessor complex, which processes the primary transcript of miR-21 (Davis *et al*, 2008). miR-21 expression in VSMCs has been associated with cell apoptosis and proliferation, and appears to be necessary for VSMC-specific gene expression (Cheng & Zhang, 2010; Albinsson & Sessa, 2011). Therefore, miR-21 might influence phenotypic changes in VSMCs from the quiescent "contractile" phenotypic state to the active "synthetic" state that can migrate and proliferate from the media to the intima during the progression of atherosclerosis (Wei *et al*, 2013).

Cumulatively, these findings demonstrate that miR-21 can regulate numerous processes involved in proper cell function, survival, and proliferation. While it has also been linked to key processes involved in inflammation, its presence is not exclusively characteristic of a pro-inflammatory or an immunosuppressive state. Instead, it functions as a key signal mediating the balance and transition between both states (Sheedy *et al*, 2010). Therefore, miR-21 induction plays an essential role in regulating the inflammatory switch. Despite these observations indicating the important role of miR-21 in regulating macrophage activation and function, the contribution of miR-21 in macrophages during the progression of atherosclerosis remains unknown.

In the current study, we define for the first time how miR-21 in hematopoietic cells impacts the progression of atherosclerosis. We show that the absence of miR-21 in hematopoietic cells accelerates the progression of atherosclerosis. Moreover, we observe that low-density lipoprotein receptor-deficient (*Ldlr*$^{-/-}$) mice transplanted with *miR-21*$^{-/-}$ bone marrow (BM) develop lesions characterized by a combination of large necrotic areas, fibrous cap thinning, and massive apoptosis. Mechanistically, we demonstrate that lack of miR-21 in macrophages increases the expression of mitogen-activated protein kinase kinase 3 (MKK3), a direct target of miR-21 (Li *et al*, 2016). This in turn leads to activation of the p38 MAP Kinase-C/EBP homologous protein (p38-CHOP) and c-Jun N-terminal kinase (JNK) signaling pathways, triggering apoptosis in endoplasmic reticulum (ER)-stressed macrophages. We also find that miR-21 plays a key role in the process of efferocytosis and the resolution of inflammation. Genetic ablation of miR-21 in macrophages attenuates clearance of apoptotic cells. Finally, we demonstrate that the absence of miR-21 reduces expression of the ATP-binding cassette transporter G1 (ABCG1), thus promoting the development of foam cell formation. These novel results dissect the role of miR-21 in hematopoietic cells during the progression of atherosclerosis and underscore the significance of miR-21 in regulating macrophage apoptosis, efferocytosis, and lipid metabolism during atherogenesis.

# Results

## miR-21 is highly expressed in monocytes/macrophages, and its absence in hematopoietic cells enhances the progression of atherosclerosis

We initially determined the expression of miRNAs expressed in primary bone marrow-derived macrophages (BMDMs) by RNA sequencing. miR-21 was highly expressed in BMDMs, accounting for 42% of total raw reads (Fig 1A). Previous miRNA profile studies in hematopoietic cells (Lazare *et al*, 2014; Teruel-Montoya *et al*, 2014) have shown that miR-21 is highly expressed in the monocytic/macrophage lineage, which is in agreement with our data (Appendix Fig S1). We next measured the expression of miR-21 in aortic lysates isolated from *Ldlr*$^{-/-}$ mice fed a Western-type diet (WD). As expected, levels of miR-21 and the macrophage marker CD68 were significantly increased in whole aortas of *Ldlr*$^{-/-}$ mice following 12 weeks of WD feeding (Fig 1B), which is likely due to the accumulation of macrophages during atherogenesis. Although other leukocyte populations have been described to play an important role during atherogenesis (Weber *et al*, 2008), monocytes are the first inflammatory cells to invade atherosclerotic lesions, and thus, macrophages become the main component of atherosclerotic plaques (Galkina & Ley, 2009). Thus, we then sorted different cell types from aortas of *Ldlr*$^{-/-}$ mice fed a WD for 12 weeks and found that the expression of miR-21 in the monocyte/macrophage

compartment was higher than all other cellular components, including neutrophils, sorted from the plaque (Fig 1C). To further confirm this, we transplanted $Ldlr^{-/-}miR-21^{-/-}$ mice with WT BM and assessed miR-21 expression after 12 weeks on WD (Aparicio-Vergara et al, 2012). Notably, in situ hybridization analysis of mouse aortic sinus plaques revealed a significant accumulation of miR-21 in CD68-positive areas of atherosclerotic plaques (Fig 1D). The specificity of this approach was confirmed by the lack of miR-21-positive cells in atherosclerotic plaques derived from $Ldlr^{-/-}miR-21^{-/-}$ mice transplanted with BM from $miR-21^{-/-}$ (Fig 1D).

To define the importance of miR-21 in hematopoietic cells during atherogenesis, we lethally irradiated $Ldlr^{-/-}$ mice and transplanted them with BM isolated from $miR-21^{-/-}$ or WT donor mice. Importantly, we found that $Ldlr^{-/-}$ mice received miR-21-deficient BM developed larger lesions than mice transplanted with WT BM (Fig 2A and B). The accelerated atherosclerosis observed in $Ldlr^{-/-}$ mice transplanted with $miR-21^{-/-}$ BM was not associated with differences in circulating lipids (Appendix Fig S2A–F).

Next, we determined the effect of miR-21 deficiency in hematopoietic cells on atherosclerotic plaque morphology. Importantly, we found a significant increase in plaque necrosis in lesions from $Ldlr^{-/-}$ mice transplanted with $miR-21^{-/-}$ BM (Fig 3A, upper panels). In contrast, fibrous caps were significantly thinner in $Ldlr^{-/-}$ mice transplanted with $miR-21^{-/-}$ BM (Fig 3A, bottom panels). The increase in plaque necrosis is associated with high rates of macrophage apoptosis. Indeed, we found a significant increase in TUNEL-positive cells in plaques isolated from animals transplanted with $miR-21^{-/-}$ BM (Fig 3B). The increased number of apoptotic cells observed in plaques from $Ldlr^{-/-}$ mice transplanted with $miR-21^{-/-}$ BM correlated with a significant reduction in macrophage content within the lesions (Fig 3C). We also examined whether the reduction in macrophage accumulation was caused by reduced monocyte/macrophage proliferation within atherosclerotic lesions. The results showed similar proliferation rates in both groups of mice, suggesting that the significant reduction in macrophage accumulation in plaques is likely due to the enhanced apoptosis observed in lesions from $Ldlr^{-/-}$ mice reconstituted with $miR-21^{-/-}$ BM (Appendix Fig S3). Finally, we also analyzed whether the absence of miR-21 in hematopoietic cells influences collagen deposition in the lesions. Surprisingly, we found that despite significant differences in fibrous cap size, the atherosclerotic plaques in both groups of mice contain similar amounts of collagen (Fig 3D). Taken together, these results demonstrate that the absence of miR-21 in hematopoietic cells accelerates the progression of atherosclerosis and promotes macrophage apoptosis, resulting in larger necrotic cores and thinner fibrous caps, both key determinants of plaque vulnerability.

## Genetic ablation of miR-21 in hematopoietic cells enhances macrophage/monocyte infiltration in the artery wall and increases the induction of inflammatory cytokines in macrophages

Increased circulating monocytes are associated with increased risk of cardiovascular disease. Thus, we assessed whether absence of miR-21 in hematopoietic cells influences blood cell counts in $Ldlr^{-/-}$ mice transplanted with $miR-21^{-/-}$ or WT BM, after feeding

a WD for 12 weeks. The results show similar proportions of circulating monocytes, neutrophils, and lymphocytes (Fig 4A). Similarly, the number of circulating neutrophils, as well as pro-inflammatory (Ly-6C$^{hi}$) and anti-inflammatory (Ly-6C$^{low}$) monocytes, was almost identical in both groups of mice (Fig 4B). While blood leukocyte counts were equivalent, we found a significant accumulation of macrophages and Ly-6C$^{hi}$ monocytes in the artery wall (Fig 4C). These data indicate that the larger plaques observed in mice transplanted with $miR-21^{-/-}$ BM cells are likely due to the overall increase in macrophage/monocyte infiltration observed in the whole aorta of this group of mice.

We then tested how the absence of miR-21 in BMDMs impacts the expression of pro- and anti-inflammatory cytokines in response to an inflammatory stimulus (LPS). Interestingly, we found that lack of miR-21 in macrophages enhances TNF-α, IL-6, and IL-1β production (Fig 4D). Moreover, COX-2 and iNOS protein levels were increased in miR-21-deficient BMDMs treated with LPS (Fig 4E).

## Absence of miR-21 in macrophages induces cell death and promotes the activation of MKK3/p38 and JNK signaling pathways

Our in vivo data demonstrate that absence of $miR-21^{-/-}$ in hematopoietic cells increases apoptosis in atherosclerotic lesions. The accumulation of apoptotic cells in the lesions is mediated by enhanced apoptotic rate or defective phagocytosis of apoptotic cells (Tabas, 2010; Moore & Tabas, 2011; Moore et al, 2013; Tabas et al, 2015). To directly test whether miR-21 plays an important role in macrophage survival, we assessed apoptosis by Annexin V staining in macrophages loaded with unesterified cholesterol [acetylated LDL (Ac-LDL) + acetyl-coenzyme A acetyltransferase (ACAT) inhibitor] or treated with the ER stress inducers thapsigargin or tunicamycin. As shown in Fig 5A, apoptosis induced by free cholesterol loading, thapsigargin, and tunicamycin was enhanced in $miR-21^{-/-}$ macrophages.

Previous studies suggested roles for MKK3/p38, JNK, and Akt in regulating macrophage apoptosis in response to ER stress (Devries-Seimon, Li et al, 2005; Seimon et al, 2009). In order to determine how miR-21 influences these signaling pathways, we treated WT and $miR-21^{-/-}$ macrophages with thapsigargin, tunicamycin, and unesterified cholesterol. The results show that absence of miR-21 might increase apoptotic susceptibility to ER stress by enhancing MKK3-p38-CHOP and JNK pathways. As shown in Fig 5B, in basal conditions, $miR-21^{-/-}$ macrophages had increased activation of p38 (increased phosphorylation), which was further induced 3 h after thapsigargin or tunicamycin treatment. Interestingly, free cholesterol loading (FC load) by incubation of macrophages with Ac-LDL plus an ACAT inhibitor produced a more pronounced and sustained activation of p38, which is agreement with previous reports (Devries-Seimon et al, 2005; Li et al, 2005). As expected, the increased activation of p38 was accompanied by an induction of CHOP (Fig 5B). The augmented activation of p38 observed in $miR-21^{-/-}$ macrophages correlates with elevated levels of MKK3 (Fig 5B), an upstream regulator of p38 and bona fide miR-21 target gene (Li et al, 2016). Moreover, we also observed a modest decrease in expression of MKP1, a phosphatase that controls p38 phosphorylation (Appendix Fig S4). In addition to p38 activation, we found an

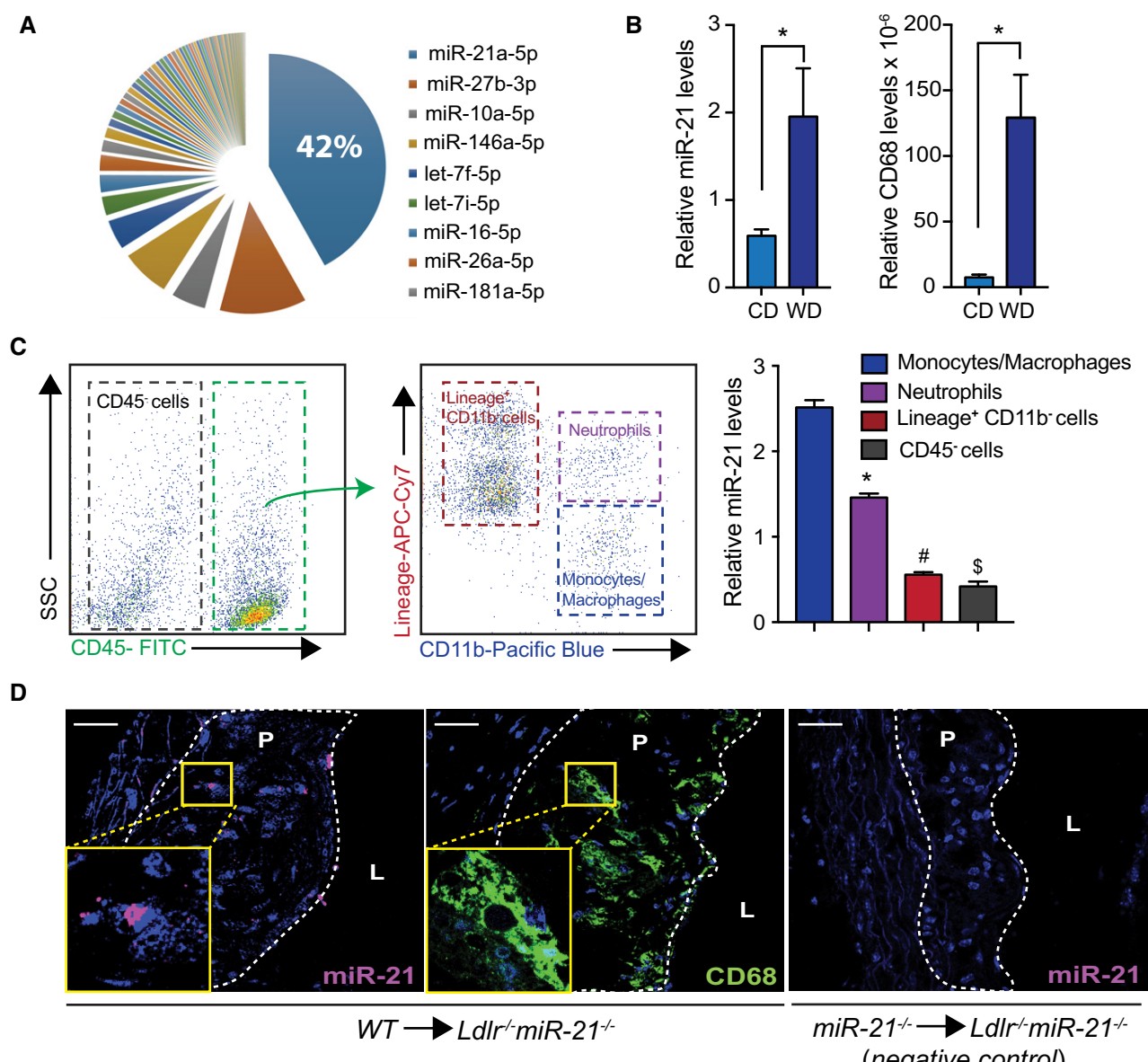

**Figure 1. miR-21 is the most abundantly expressed miRNA in macrophages.**

A   Pie graph representation of miRNAs in mouse BMDMs; top nine most abundant are indicated. Small RNA-seq analysis of BMDM cellular miRNA reads. Data are expressed as average reads per million (RpM) which are reads mapped to a miRNA per total mapped reads across samples (*n* = 3).

B   Relative miR-21 (left) or CD68 macrophage marker (right) expression in whole aorta of *Ldlr*$^{-/-}$ mice fed a chow diet (CD) or after feeding Western-type high-fat diet (WD) for 12 weeks. miR-21 levels are normalized to U6, and CD68 levels are normalized to 18S rRNA. Data represent the mean ± SEM (*n* = 5 per group; *\*P* = 0.048 for mR-21 expression and *\*P* = 0.01 for CD68 expression). Level of significance was determined using *t*-test.

C   Left, representative contour plots demonstrate the gating scheme for sorting aortic cells. Monocytes/macrophages (CD11b$^+$/lineage$^-$), neutrophils (CD11b$^+$/lineage$^+$), lineage$^+$ cells are CD45$^+$ cells not including monocytes/macrophages and neutrophils, and CD45$^-$ cells are aortic non-leukocytic cells. Right, relative miR-21 expression levels normalized to U6 of different cell types sorted from whole aortas of *Ldlr*$^{-/-}$ mice fed a WD for 12 weeks. Data represent the mean ± SEM (*n* = 3; *\*,#,\$P* < 0.0001, compared to monocytes/macrophages). Level of significance was determined using one-way ANOVA with Bonferroni's post-test.

D   Representative *in situ* hybridization of miR-21 (left) in atherosclerotic plaques isolated from double-knockout (DKO) *Ldlr*$^{-/-}$*miR-21*$^{-/-}$ mice transplanted with WT bone marrow (BM) and fed a WD for 12 weeks. Image in the middle is a representative staining for CD68 in a consecutive section to the one used for *in situ* hybridization. The image on the right shows a negative control for detection of miR-21 in plaque macrophages of DKO mice transplanted with *miR-21*$^{-/-}$ BM. P, plaque. L, lumen. Scale bar, 50 μm.

enhanced activation of JNK in *miR-21*$^{-/-}$-deficient macrophages. Interestingly, the increase in JNK phosphorylation was independent of PDCD4 expression (Fig 5B), a negative regulator of JNK

activation and well-established miR-21 target (Asangani *et al*, 2008; Frankel *et al*, 2008). Together, these results suggest that the ER-stress-induced macrophage apoptosis observed in

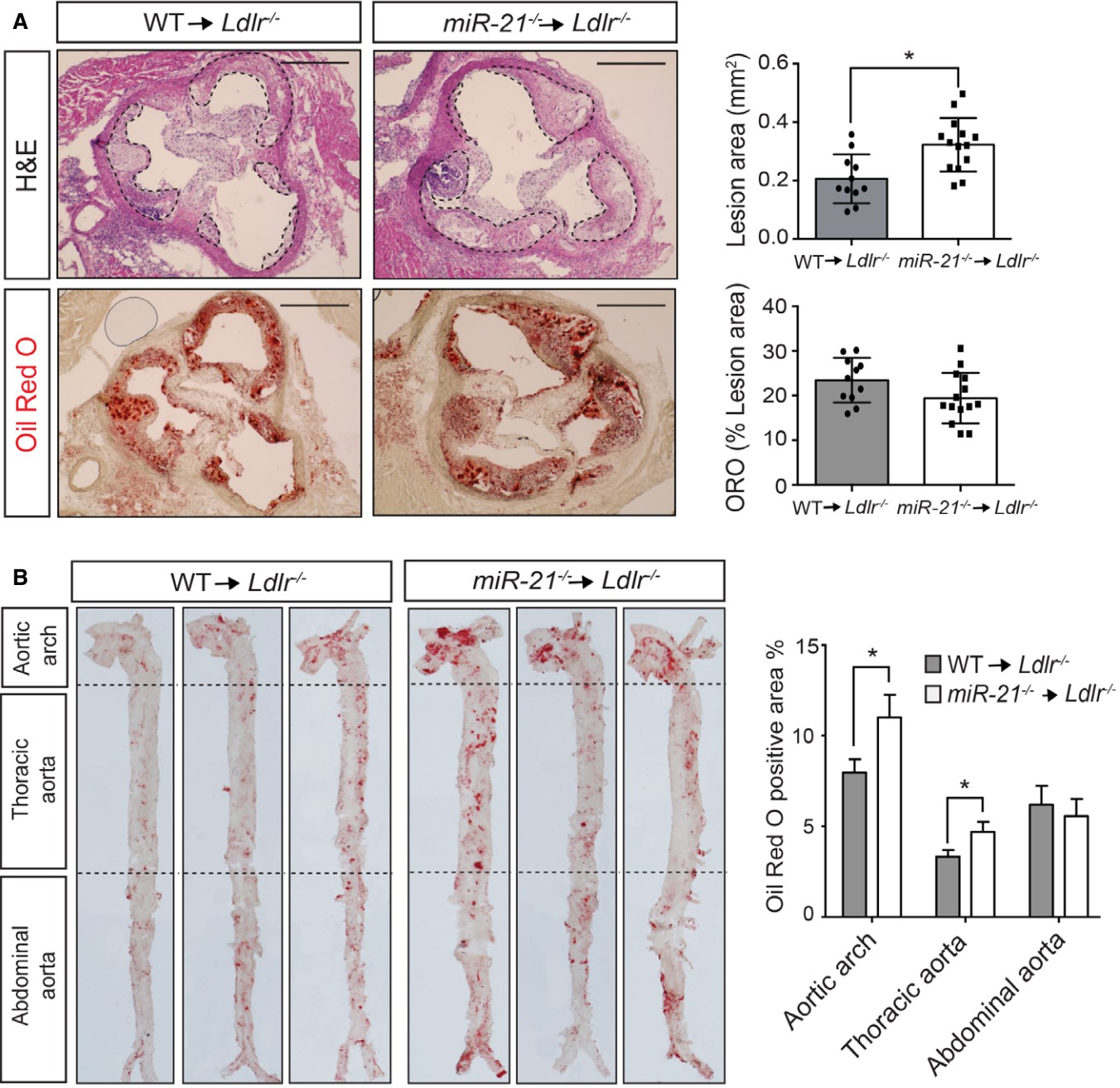

**Figure 2. miR-21 absence in macrophages enhances atherogenesis.**

A Representative histological analysis of cross sections of the aortic sinus stained with hematoxylin and eosin (H&E) or Oil Red O (ORO) of $Ldlr^{-/-}$ mice transplanted with WT or $miR-21^{-/-}$ BM fed for 12 weeks on a WD (left). Right panels are the mean lesion area calculated from H&E or ORO aortic cross sections, respectively. Each dot represents the mean of the quantification of nine sections from an individual animal. Data represent the mean ± SEM ($n$ = 11–14 per group; *$P$ = 0.0051). Scale bars, 400 μm. Level of significance was determined using Mann–Whitney test.

B Representative ORO staining of three aortas of $Ldlr^{-/-}$ mice transplanted with WT or $miR-21^{-/-}$ BM fed for 12 weeks on a WD (left). Quantification of the ORO-positive area is shown in the right panel. Data represent the mean ± SEM ($n$ = 9–11 per group; *$P$ = 0.0465 for the aortic arch and *$P$ = 0.0310 for the thoracic aorta). Level of significance was determined using one-way ANOVA with Bonferroni's post-test.

miR-21-deficient macrophages correlates with increased activation of p38 and JNK pathways, likely due to de-repression of MKK3.

We also analyzed the expression of other described miR-21 target genes that regulate cellular proliferation and survival such as PTEN.

Previous reports have shown that miR-21 inhibits PTEN expression (Meng *et al*, 2007; Perdiguero *et al*, 2011). PTEN is a phosphatase that activates the phosphatidylinositol-4,5-bisphosphate 3-kinase (PI3K)/AKT pathway, thus promoting cellular proliferation and

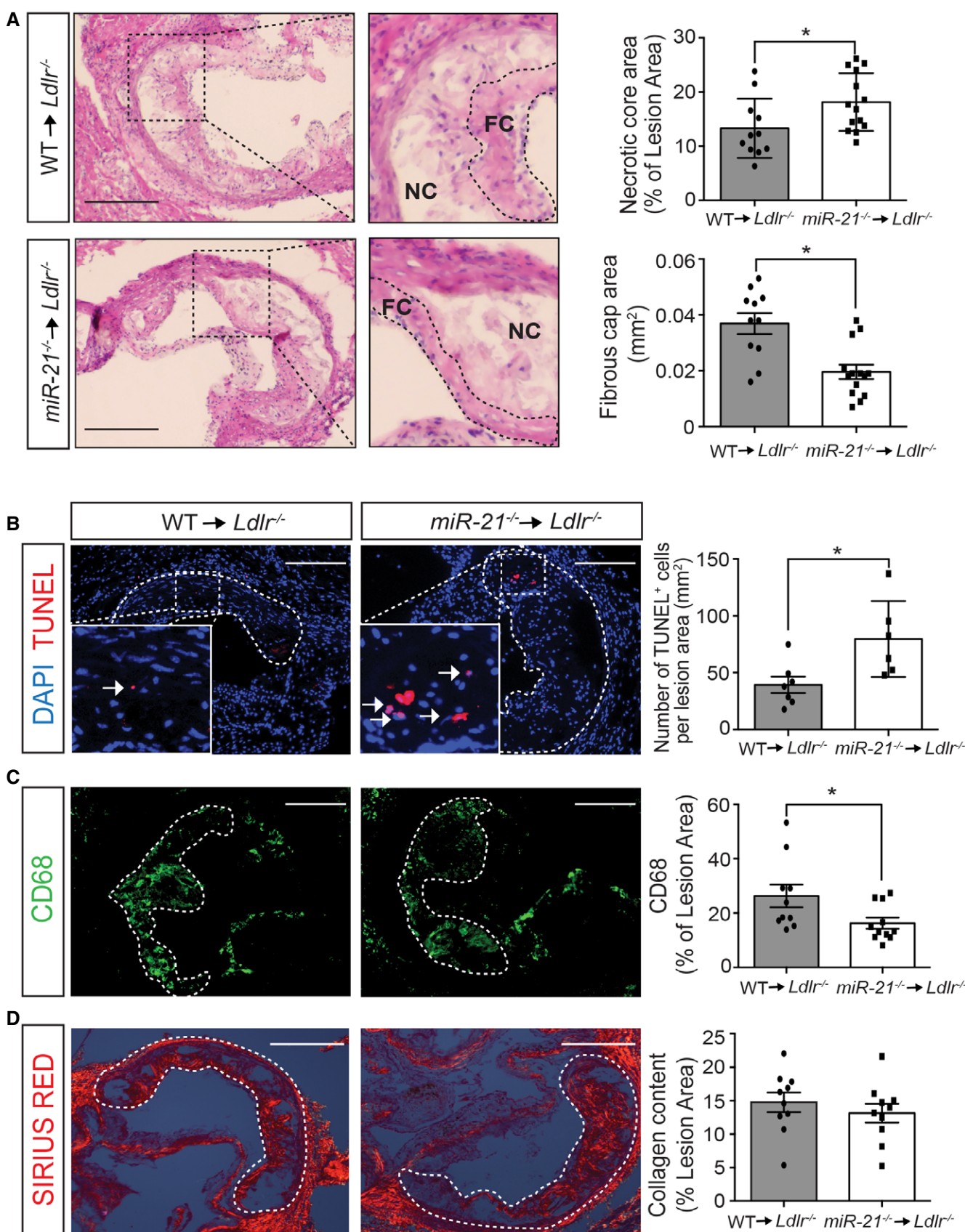

**Figure 3.**

**Figure 3.  Increased macrophage apoptosis and plaque necrosis in *Ldlr*$^{-/-}$ mice transplanted with *miR-21*$^{-/-}$ BM.**

A    Representative images of aortic root cross sections stained with hematoxylin and eosin (H&E) from *Ldlr*$^{-/-}$ mice transplanted with WT or *miR-21*$^{-/-}$ BM fed for 12 weeks on WD. Enlarged images shown in the right panels reveal how the necrotic core area was defined for quantification in each section and the area of the fibrous cap that was used to measure the thickness. Dashed lines show the boundary of the developing necrotic core (NC) and fibrous cap (FC), see Materials and Methods section for a detailed description. Quantification of the necrotic core area and fibrous cap is shown in the right panels. Each dot represents the mean of the quantification of three sections from individual animals. Data represent the mean ± SEM (*n* = 11–14 per group, as indicated; *P = 0.0179 for necrotic core and *P = 0.001 for cap thickness). Scale bars, 200 μm. Level of significance was determined using Mann–Whitney test.

B–D  Representative cross-sectional analysis of the aortic root of *Ldlr*$^{-/-}$ mice transplanted with WT or *miR-21*$^{-/-}$ BM fed for 12 weeks on WD (B) for apoptotic cell analysis via TUNEL staining, (C) macrophage content analysis via immunodetection of CD68, and (D) collagen content analysis via Sirius Red staining. Quantifications are the graphs on the right. Each dot represents the mean of the quantification of three sections from individual animals. Data represent the mean ± SEM (*n* = 6–11 per group, as indicated). (B) Data are represented as the number of TUNEL-positive cells per lesion area; *P = 0.014. (C) Data are expressed as the average percentage of CD68-positive signal per total lesion area; *P = 0.0159. (D) Data are expressed as the average percent collagen per total lesion area. Scale bars, 200 μm. Level of significance was determined using Mann–Whitney test.

survival. Surprisingly, we found that absence of miR-21 in macrophages does not influence PTEN expression levels (Appendix Fig S4). However, we found a modest increase in AKT phosphorylation in macrophages treated with tunicamycin and thapsigargin (Appendix Fig S4), a finding that could be explained by the significant activation of p38 observed in miR-21-deficient macrophages.

To further define the effect of miR-21 on the MKK3-p38-CHOP signaling pathway, we transfected mouse primary macrophages with control miRNA mimics (CM) or a miR-21 mimic and assessed MKK3 and p38 phosphorylation. As expected, overexpression of miR-21 markedly reduced the thapsigargin-induced activation of MKK3. As such, we also observed a reduction in p38 phosphorylation (Fig 5C).

Next, we determined the contribution of p38 activation on the induction of CHOP by treating WT and *miR-21*$^{-/-}$ macrophages with SB203580, a p38 inhibitor. Interestingly, we found that WT and *miR-21*$^{-/-}$ macrophages treated with SB203580 had similar CHOP levels, suggesting that the increase in CHOP observed in *miR-21*$^{-/-}$ macrophages compared to WT macrophages was likely mediated by p38 activation (Fig 5D). Together, these results demonstrate that the absence of miR-21 activates the pro-apoptotic MKK3-p38-CHOP and JNK signaling pathways.

## miR-21 deficiency in macrophages attenuates apoptotic clearance associated with reduced MERTK protein levels

Next, we determined whether the absence of miR-21 influences apoptotic cell clearance, thus promoting the accumulation of apoptotic cells in atherosclerotic lesions isolated from *Ldlr*$^{-/-}$ mice transplanted with *miR-21*$^{-/-}$ BM (Fig 3B). To this end, we incubated mouse peritoneal macrophages isolated from WT and *miR-21*$^{-/-}$ mice with apoptotic Jurkat cells labeled with CellTracker Red, followed by removal of non-ingested cells by vigorous washing. Then, we analyzed the engulfment of apoptotic cells by macrophages using confocal microscopy. Interestingly, we found that apoptotic cells were efficiently engulfed by WT macrophages, whereas phagocytosis by *miR-21*$^{-/-}$ macrophages was markedly reduced (Fig 6A, quantification right panel).

To identify the potential mechanisms associated with the deficient removal of apoptotic cells, we assessed whether miR-21 influences the expression of MERTK, a key receptor that mediates the clearance of apoptotic cells in atherosclerotic lesions of hypercholesterolemic mice (Thorp *et al*, 2008, 2011). Of note, we found that total MERTK levels were significantly reduced in miR-21-deficient macrophages compared to WT macrophages (Fig 6B). Similar

results were observed when we activated the cleavage of MERTK by incubating macrophages in the presence of LPS or when we activated the transcription of MERTK by treating cells with a synthetic LXR agonist (T0901317; Fig 6B). We also assessed the cell membrane expression MERTK in WT and *miR-21*$^{-/-}$ macrophages by flow cytometry and found a significant reduction of MERTK expression in both untreated and LPS-treated macrophages (Fig 6C). Cellular MERTK levels were not affected by increasing MERTK cleavage, as the ratio of soluble MERTK (sMERTK) vs. cellular MERTK was similar in WT and *miR-21*$^{-/-}$ macrophages (Fig 6D). Furthermore, the absence of miR-21 did not influence *Mertk* mRNA levels (Fig 6E). Taken together, these results suggest that miR-21 affects MERTK expression at post-transcriptional level but independent of proteolytic processing.

## Macrophage miR-21 deficiency enhances ABCG1 degradation and increases foam cell formation

We next examined whether miR-21 also regulates cholesterol metabolism in macrophages, an important event in the early stages of atherosclerotic lesions (Lusis, 2000; Glass & Witztum, 2001). To this end, we incubated WT and *miR-21*$^{-/-}$ macrophages in the presence of acetylated LDL (Ac-LDL) for 24 h and assessed neutral lipid accumulation by staining cells with Oil Red O (ORO). Of note, we found a significant increase in lipid accumulation in macrophages isolated from *miR-21*$^{-/-}$ mice compared to WT macrophages (Fig 7A, quantification in right panels). Similarly, we observed an increase in the total cholesterol of miR-21-deficient macrophages compared to WT macrophages (Fig 7B). To determine the molecular mechanisms that could contribute to the increased lipid content of *miR-21*$^{-/-}$ macrophages, we assessed how miR-21 influences lipoprotein uptake and cholesterol efflux. Flow cytometry analysis revealed no differences in the uptake of DiI-labeled ox-LDL (DiI-Ox-LDL) between WT and *miR-21*$^{-/-}$ macrophages (Fig 7C). However, we observed a significant decrease in cholesterol efflux to high-density lipoproteins (HDL) in *miR-21*$^{-/-}$ macrophages compared to WT (Fig 7D). Importantly, the defective cholesterol efflux observed in *miR-21*$^{-/-}$ macrophages correlates with a significant reduction in ABCG1 expression, a transporter that regulates the efflux of cholesterol to HDL (Fig 7E). The expression of ABCA1, a transporter responsible for cholesterol efflux to poorly lipidated ApoA1, was similar in both groups of macrophages (Fig 7E). Previous reports have shown that p38 activation stimulates ABCG1 phosphorylation and degradation (Nagelin *et al*, 2009). To define the role of the p38 signaling pathway in regulating ABCG1 expression, we treated WT

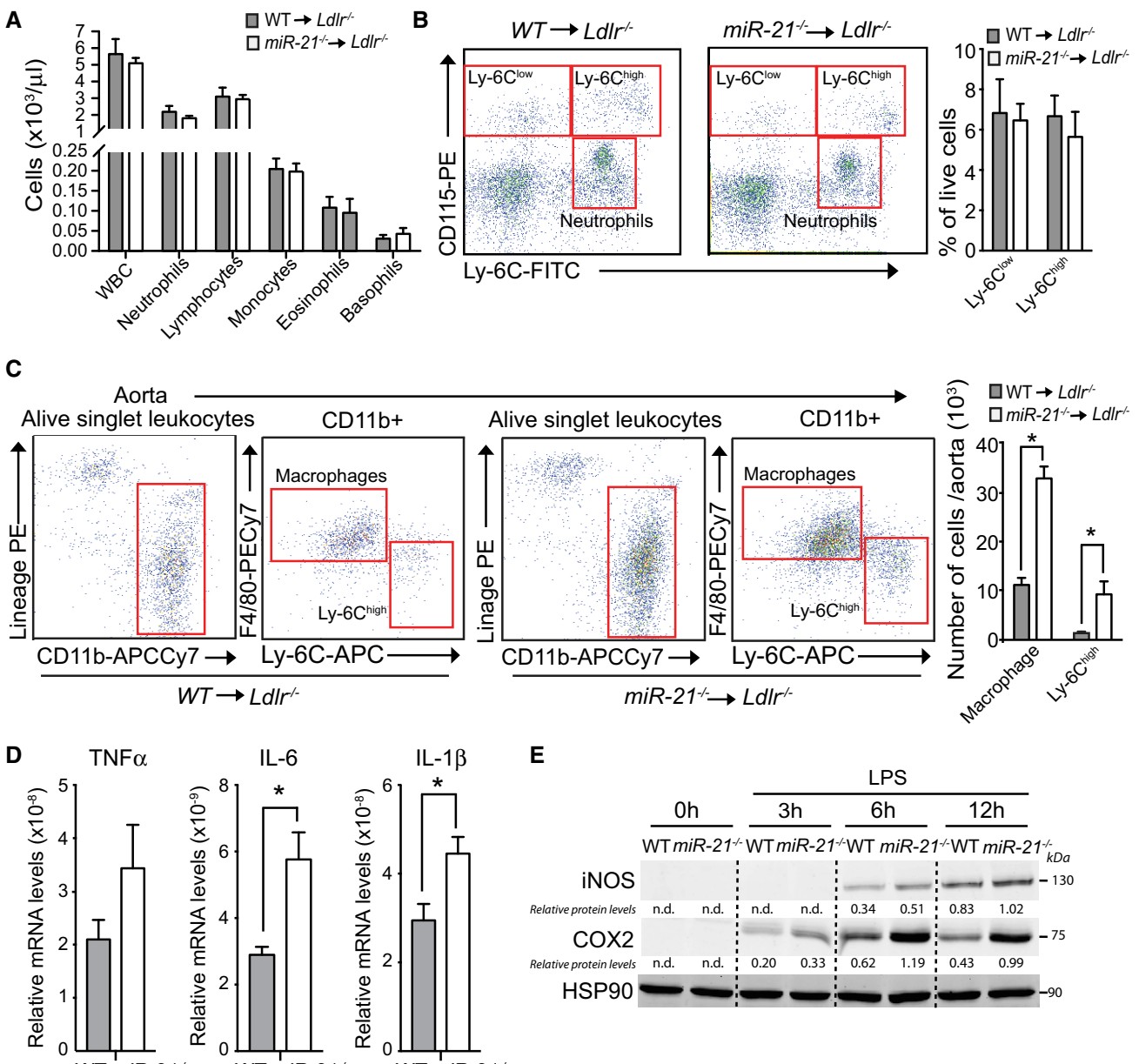

**Figure 4. Aortas of *Ldlr*<sup>−/−</sup> mice transplanted with *miR-21*<sup>−/−</sup> BM have increased numbers of pro-inflammatory macrophages, and miR-21 deficiency increases the expression of inflammatory cytokines.**

A   Peripheral blood counts from *Ldlr*<sup>−/−</sup> mice transplanted with WT or *miR-21*<sup>−/−</sup> BM after 12 weeks on WD measured using Hemavet hematology analyzer. Data represent the mean ± SD (n = 11 per group).

B   Flow cytometry analysis of circulating monocytes and neutrophils from *Ldlr*<sup>−/−</sup> mice transplanted with WT or *miR-21*<sup>−/−</sup> BM fed for 12 weeks on WD. Left, representative dot plots showing gaiting schemes. Right, quantification Ly-6C<sup>low</sup> and Ly-6C<sup>high</sup> circulating monocytes expressed as % of live cells. Data represent the mean ± SEM (n = 6 for WT and n = 5 for *miR-21*<sup>−/−</sup>).

C   Flow cytometry analysis of aortic macrophages and monocytes from *Ldlr*<sup>−/−</sup> mice transplanted with WT or *miR-21*<sup>−/−</sup> BM fed for 12 weeks on a WD. Left, representative dot plots showing gating schemes. Right, quantification of total number of macrophages or Ly-6C<sup>high</sup> monocytes per aorta. Data represent the mean ± SEM (n = 3 per group; *P = 0.0001 for macrophages and *P = 0.004 for Ly-6C<sup>high</sup> monocytes). Level of significance was determined using one-way ANOVA with Bonferroni's post-test.

D   Relative TNF-α, IL-6, or IL-1β mRNA expression levels normalized to 18S rRNA of WT or *miR-21*<sup>−/−</sup> BMDMs treated with LPS (10 ng/ml) for 6 h. Data are mean ± SEM (n = 4 per group; *P = 0.0152 for IL-6 and *P = 0.0291 for IL-1β). Level of significance was determined using *t*-test.

E   Western blot analysis of iNOS and COX-2 in WT or *miR-21*<sup>−/−</sup> BMDMs treated with LPS (10 ng/ml) for the indicated times. HSP90 was used as a loading control. Representative experiment out of three with similar results. Relative protein levels were determined by band densitometry and are expressed in arbitrary units after correction for HSP90. n.d., not detectable.

Source data are available online for this figure.

and $miR\text{-}21^{-/-}$ macrophages with SB203580 and assessed ABCG1 levels by Western blot. The results show that inhibition of p38 increases ABCG1 protein levels in miR-21-deficient macrophages, suggesting that the constitutive basal p38 activation found in $miR\text{-}21^{-/-}$ macrophages influences ABCG1 levels (Fig 7F). Finally, we assessed whether the inhibition of the proteasome enhances the expression of ABCG1 in miR-21-deficient macrophages. To this end, we incubated WT and $miR\text{-}21^{-/-}$ macrophages in the presence of Ac-LDL (to induce ABCG1 expression) and treated them with MG132, a proteasome inhibitor. As seen in Fig 7G, ABCG1 protein levels were similar in WT and $miR\text{-}21^{-/-}$ macrophages, suggesting that enhanced proteolytic activity in macrophages lacking miR-21 could explain the reduced ABCG1 expression in macrophages.

## Discussion

The most salient finding of this study is that the absence of miR-21 in hematopoietic cells enhances the progression of atherosclerosis and promotes adverse plaque remodeling, characterized by increased necrosis and fibrous cap thinning in atherosclerotic lesions. The accelerated atherogenesis observed in hematopoietic miR-21-deficient mice seems to have resulted from a combination of distinct mechanisms including (i) an increased pro-inflammatory phenotype, (ii) enhanced macrophage apoptosis and foam cell formation, and (iii) decreased macrophage phagocytic capacity.

Several lines of evidence indicate that atherosclerotic progression depends on persistent and chronic inflammation in the arterial walls. miR-21 is involved in cardiovascular disease and has been described as a key signal mediating the balance of pro- and anti-inflammatory responses (Sheedy, 2015). Importantly, overexpression of miR-21 results in a significant suppression of LPS-induced TNF-α expression and NF-ĸB activation. In agreement with these observations, we found that miR-21-deficient macrophages have increased levels of PDCD4 and elevated LPS-induced expression of TNF-α, IL-6, and IL-1β. Moreover, in line with the pro-inflammatory phenotype observed in $miR\text{-}21^{-/-}$ macrophages, we also observed increased levels of COX-2 after LPS stimulation. Interestingly, the report by Sheedy (2015) indicates that miR-21 targeting of PDCD4 is associated with increased IL-10 production, an anti-inflammatory

cytokine associated with resolution of inflammation. Additionally, the digestion of dead cells by phagocytes is often associated with the induction of anti-inflammatory cytokines such as IL-10 (Sheedy, 2015). However, other reports have shown that brief exposure to free cholesterol-induced apoptotic macrophages can induce the production of pro-inflammatory cytokines such as TNF-α and IL-1β (Sheedy, 2015). Recently, it has been shown that miR-21 plays an important role in the regulation of efferocytosis-mediated suppression of innate immune response, a key process implicated in resolving inflammation following injury (Das et al, 2014). Accumulation of apoptotic macrophages in late atherosclerotic lesions, caused by an enhanced apoptotic susceptibility or defective phagocytosis of apoptotic cells, results in increased plaque necrosis and a heightened state of inflammation in the vessel wall (Tabas, 2010; Tabas et al, 2015). Das et al (2014) demonstrate that miR-21 levels are induced in response to LPS via PDCD4 repression. Altogether, these data indicate that absence of miR-21 promotes a pro-inflammatory and anti-resolution phenotype and suggest that miR-21 plays a key role during the resolution of inflammation, an essential process that limits the progression and promotes the regression of atherosclerosis.

Numerous studies have documented that activation of p38 MAPK can have pro- or anti-apoptotic effects depending on the cellular environment. Early observations by the Tabas laboratory demonstrated that p38 signaling was necessary for CHOP induction and apoptosis in macrophages loaded with cholesterol (Devries-Seimon et al, 2005). In addition to p38, the authors also identified JNK as an important pathway associated with apoptosis, but independent of CHOP (Devries-Seimon et al, 2005). Similar to these findings, we found that the absence of miR-21 increases p38 and JNK signaling cascades and promotes apoptosis in response to ER stress in vitro. To determine how miR-21 influences p38 phosphorylation, we assessed the expression of MKK3, an established miR-21 target gene that regulates p38 phosphorylation (Li et al, 2016). Importantly, we found a marked increase in MMK3 expression in $miR\text{-}21^{-/-}$ macrophages compared to WT macrophages. Similarly, Tabas et al observed that p38 phosphorylation in response to cholesterol overloading was blunted in MKK3-deficient macrophages (Li et al, 2005). While these results suggest that activation of p38 via MKK3 might attenuate macrophage apoptosis in vitro and reduce plaque

---

**Figure 5.   Absence of miR-21 enhances macrophage apoptosis and activates MKK3-p38-CHOP and JNK signaling pathways.**

A   Determination of apoptosis by flow cytometry of peritoneal macrophages isolated from WT or $mir\text{-}21^{-/-}$ mice treated with thapsigargin (2 µM), tunicamycin (5 µg/ml), or loaded with free cholesterol (FC load) by treatment with Ac-LDL (120 µg cholesterol/ml) + ACAT inhibitor [Sandoz 58035 (10 µg/ml)] for 24 h. Left, dot plots showing gating schemes of one representative experiment. Right, data are average of % of non-viable cells (blue), which are the sum of late apoptotic cells (red, annexin V⁺/PI⁺) and the early apoptotic cells (green, annexin V⁺/PI⁻). Data represent the mean ± SEM (n = 5 for WT and n = 6 for $miR\text{-}21^{-/-}$; *P = 0.003 for thapsigargin treatment, *P = 0.04 for tunicamycin treatment, and *P = 0.003 for FC load). Level of significance was determined using one-way ANOVA with Bonferroni's post-test.

B   Western blot analysis of CHOP, p-p38, p38, p-JNK, JNK, p-MKK3, MKK3, and PDCD4 in WT or $miR\text{-}21^{-/-}$ peritoneal macrophages treated with thapsigargin (2 µM), tunicamycin (5 µg/ml), or FC load with Ac-LDL (120 µg cholesterol/ml) + ACAT inhibitor (10 µg/ml) for the indicated times. Relative protein levels were determined by band densitometry and are expressed in arbitrary units after correction for HSP90 (loading control). n.d., not detectable.

C   Western blot analysis of pMKK3, MKK3, p38, p-p38, and CHOP in WT peritoneal macrophages transfected with control miRNA mimic (CM) or miR-21 mimic and treated or not with thapsigargin (2 µM) for 6 h. Relative protein levels were determined by band densitometry and are expressed in arbitrary units after correction for HSP90 (loading control). n.d., not detectable.

D   Western blot analysis of p38, p-p38, and CHOP expression in WT or $miR\text{-}21^{-/-}$ peritoneal macrophages treated with SB202190 (10 µM) for 2 h prior treatment with thapsigargin (2 µM) for the indicated times. Relative protein levels were determined by band densitometry and are expressed in arbitrary units after correction for HSP90 (loading control). n.d., not detectable.

Source data are available online for this figure.

necrosis *in vivo*, other reports have demonstrated a pro-atherogenic role of p38 activation depending upon the apoptotic stimulus (Devries-Seimon *et al*, 2005; Seimon *et al*, 2009). In this regard, p38

inhibition combined with ER stress and activation of pattern recognition receptors triggers macrophage apoptosis *in vitro*. Mechanistically, the authors observed that genetic ablation or pharmacological

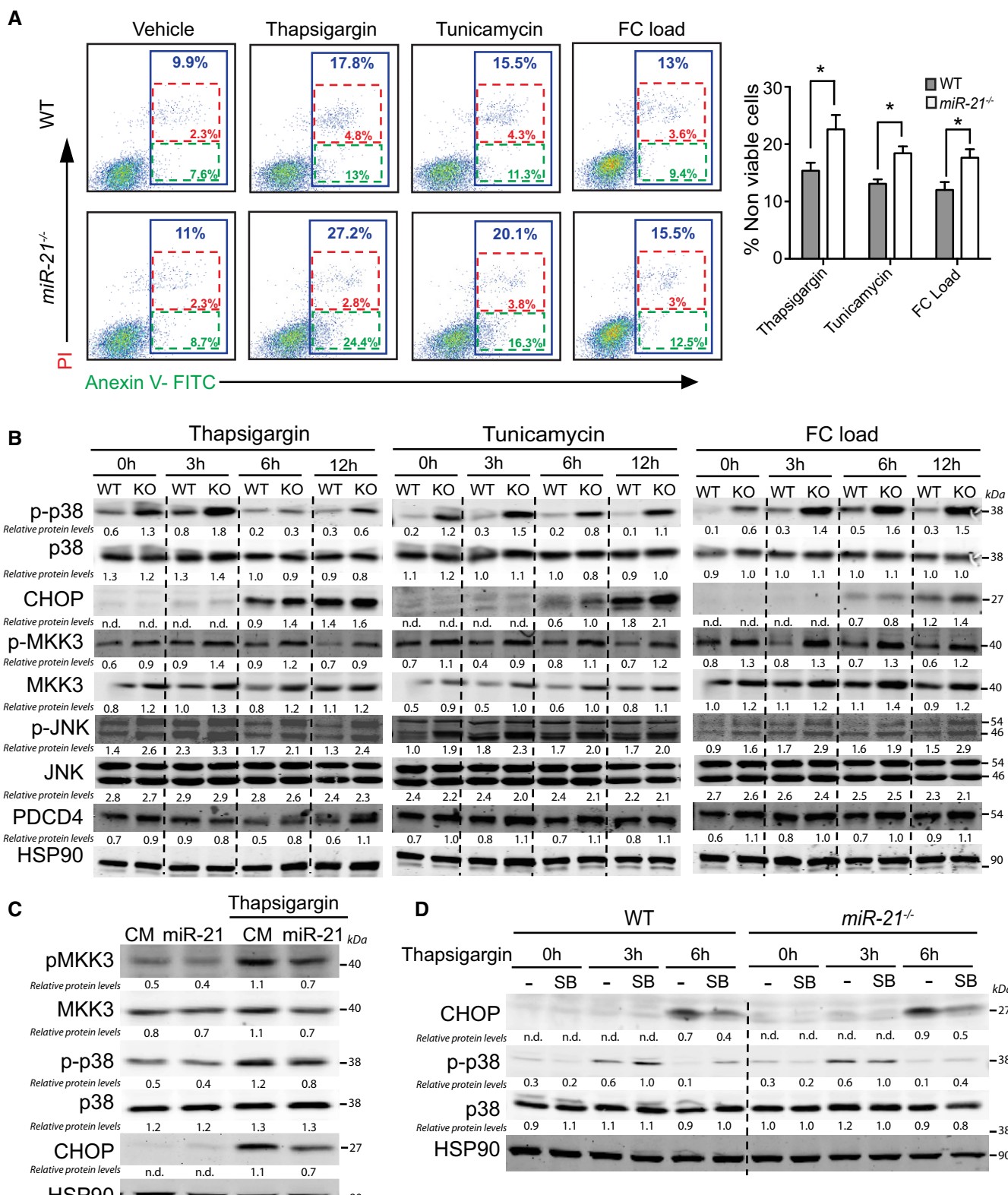

**Figure 5.**

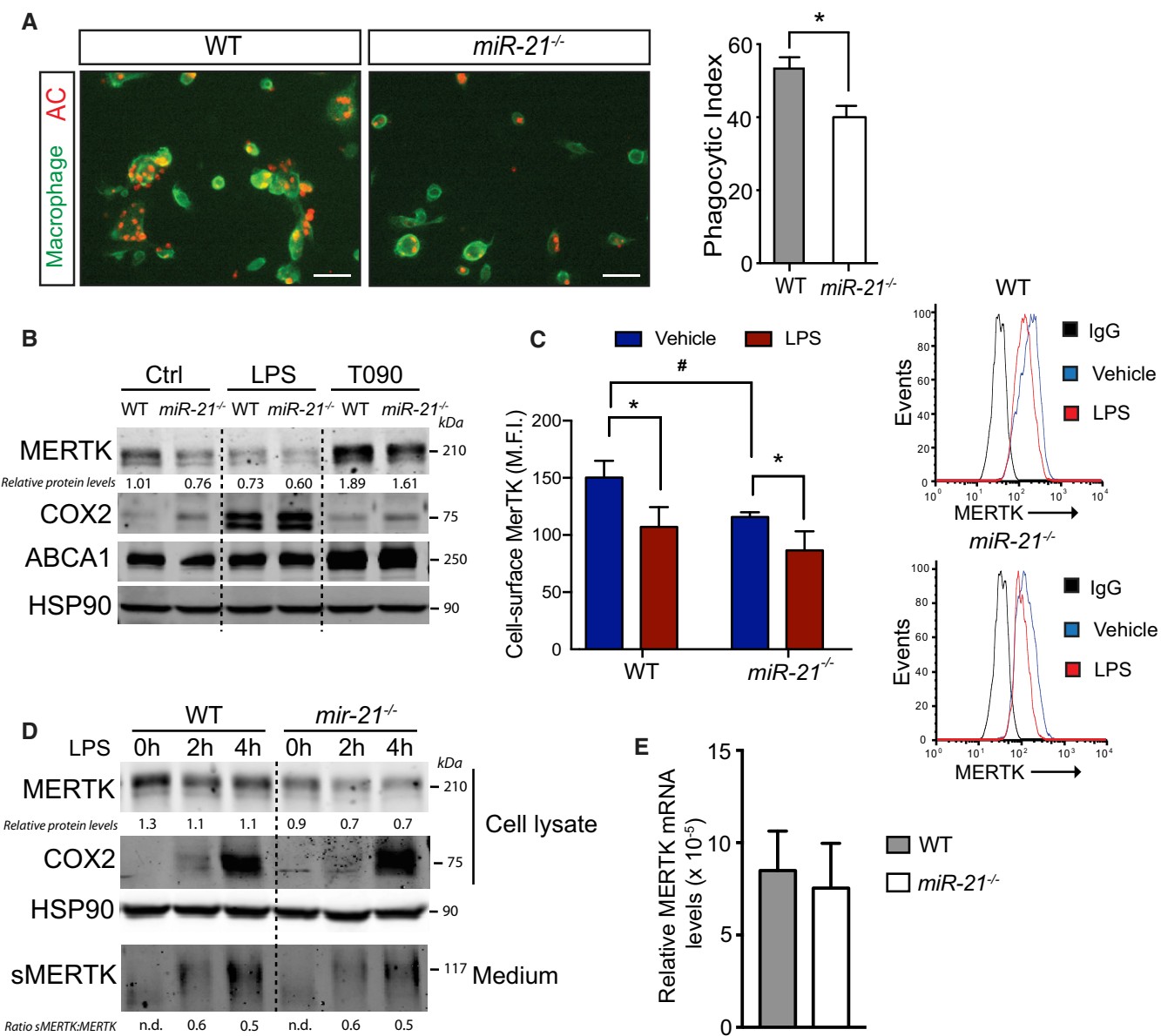

**Figure 6.  miR-21 deficiency diminishes efferocytosis and decreases MERTK expression in macrophages.**

A  Representative confocal microscopy images (left) of the *in vitro* engulfment of CellTracker Red labeled apoptotic Jurkat cells by peritoneal macrophages isolated from WT or *miR-21*[−/−] mice. Right, phagocytosis expressed as phagocytic index, which is the number of apoptotic cells (red) ingested in 1 h per F4/80-positive macrophage (green) × 100. Data represent the mean ± SEM of the quantification of five images from different fields of duplicate samples (*n* = 3 per group; *P = 0.017). Scale bars, 100 μm. Level of significance was determined using *t*-test.

B  Representative Western blot analysis of MERTK protein levels in WT or *miR-21*[−/−] peritoneal macrophages treated with LPS (50 ng/ml; to induce MERTK cleavage) and T0901317 (3 μM; to induce MERTK expression) for 12 h. COX2 and ABCA1 are positive controls for LPS or T0901317 treatment, respectively. Representative experiments out of three with similar results. Relative protein levels were determined by band densitometry and are expressed in arbitrary units after correction for HSP90 (loading control).

C  Flow cytometry analysis of cell surface MERTK expression in WT and *miR-21*[−/−] macrophages treated or nor with LPS (50 ng/ml) for 12 h. The results are expressed in terms of geometric mean fluorescence intensity (M.F.I.) after subtracting isotype control (IgG). Histograms of one representative experiment are shown on the right. Data represent the mean ± SEM (*n* = 3 per group; #P = 0.009 WT vs. *miR-21*[−/−] macrophages vehicle; *P = 0.0012 vehicle vs. LPS treatment in WT macrophages and *P = 0.032 for vehicle vs. LPS treatment in *miR-21*[−/−] macrophages). Level of significance was determined using one-way ANOVA with Bonferroni's post-test.

D  Representative Western blot analysis of cellular MERTK and secreted MERTK (sMERTK) in WT and *miR-21*[−/−] peritoneal macrophages treated with LPS (10 ng/ml) for 2 and 4 h. COX2 is used a positive control for LPS treatment. Representative experiments out of three with similar results. Relative protein levels were determined by band densitometry and are expressed in arbitrary units after correction for HSP90 (loading control). n.d., not detectable.

E  Relative MERTK mRNA expression levels normalized to 18S rRNA of peritoneal macrophages isolated from WT and *miR-21*[−/−] mice. Data are mean ± SEM (*n* = 3 per group).

Source data are available online for this figure.

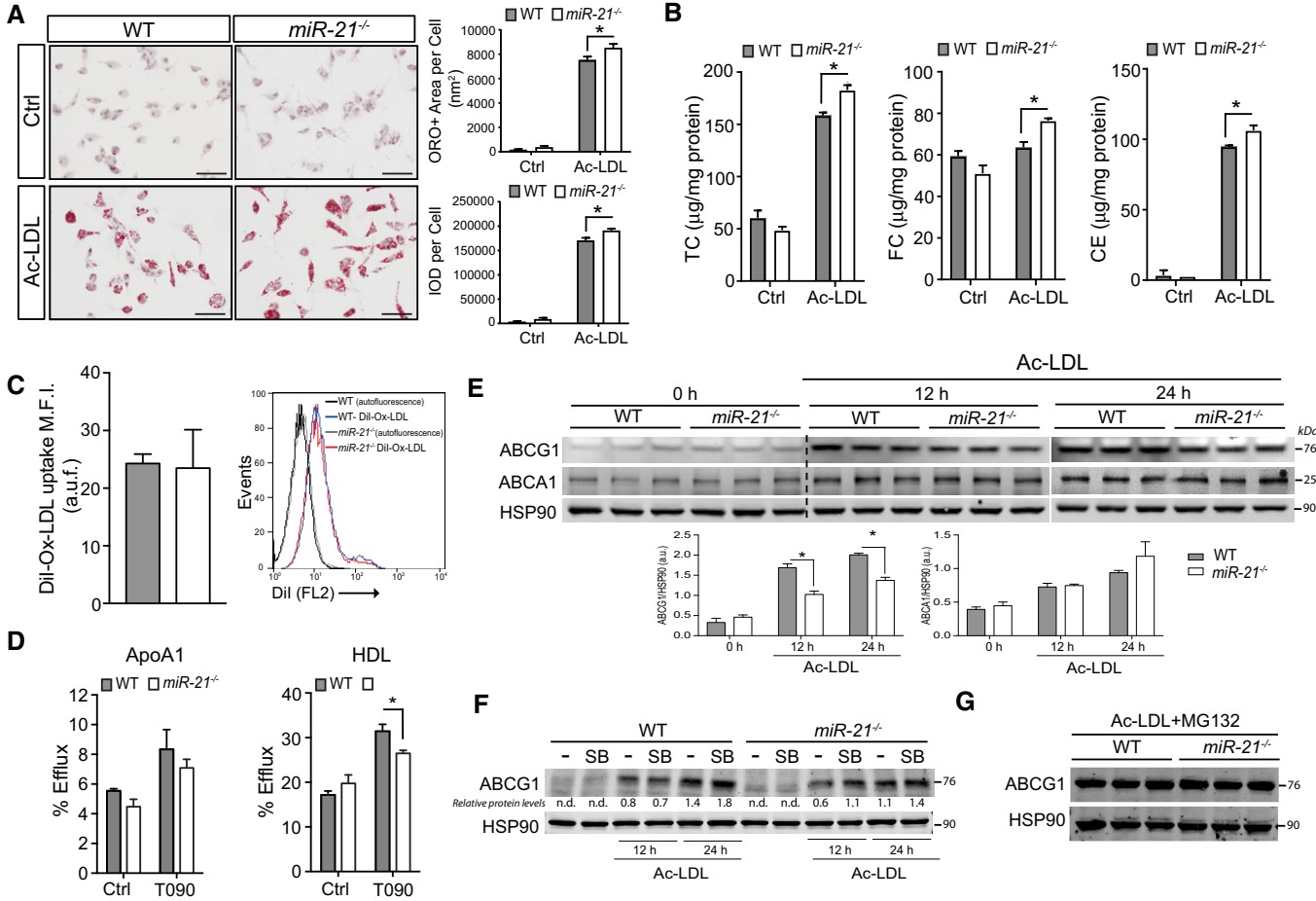

**Figure 7. Absence of miR-21 in macrophages promotes foam cell formation.**

A   Representative images of WT or *miR-21^−/−* peritoneal macrophages incubated with or without Ac-LDL (120 μg cholesterol/ml) for 24 h and stained with ORO. Quantification of the ORO-positive area and integrated optical density (IOD) per cell is shown in the right panels. Data represent the mean ± SEM of the quantification of three images from different fields (*n* = 3 per group; *P = 0.0126 for ORO and *P = 0.0148 for IOD). Scale bars, 100 μm. Level of significance was determined using one-way ANOVA with Bonferroni's post-test.

B   Quantification of total cholesterol (TC) (left panel), free cholesterol (FC) (middle panel), and cholesterol esters (CE) (right panel) corrected by cell protein concentration of WT and *miR-21^−/−* peritoneal macrophages incubated as in (A). Data represent the mean ± SEM of duplicates (*n* = 3 per group; *P = 0.022 for TC, *P = 0.042 for FC and *P = 0.025 for CE). Level of significance was determined using one-way ANOVA with Bonferroni's post-test.

C   Flow cytometry analysis of DiI-Ox-LDL uptake in peritoneal macrophages of WT and *miR-21^−/−* mice incubated with DiI-Ox-LDL (30 μg cholesterol/ml) for 2 h at 37°C. The results are expressed in terms of geometric mean fluorescence intensity (M.F.I.) after subtracting the autofluorescence of cells incubated in the absence of DiI-Ox-LDL. Data represent the mean ± SEM of duplicates (*n* = 3 per group)

D   Cholesterol efflux to apolipoprotein A1 (ApoA1) and HDL in peritoneal macrophages isolated from WT and *miR-21^−/−* mice stimulated with or without T0901317 (T090). Data represent the mean ± SEM of duplicate samples (*n* = 3 per group; *P = 0.0417). Level of significance was determined using *t*-test.

E   Representative Western blot analysis of ABCG1 and ABCA1 in WT and *miR-21^−/−* peritoneal macrophages treated with Ac-LDL (120 μg/ml) for 12 and 24 h. Samples from three different mice per group are shown. Bottom panels show the quantification of band densitometry values of ABCG1 and ABCA1 protein levels expressed in arbitrary units after correction for HSP90 (loading control). Data represent the mean ± SD from blots above (*n* = 3 per group as indicated; *P = 0.0002 for 12 h and *P = 0.0003 for 24 h). Level of significance was determined using one-way ANOVA with Bonferroni's post-test.

F   Representative Western blot analysis of ABCG1 and ABCA1 of WT and *miR-21^−/−* peritoneal macrophages treated with Ac-LDL (120 μg/ml) in the presence of p38 inhibitor (SB202190) for 12 and 24 h. Representative experiments out of three with similar results. Relative protein levels were determined by band densitometry and are expressed in arbitrary units after correction for HSP90 (loading control). n.d., not detectable.

G   Representative Western blot analysis of ABCG1 in WT and *miR-21^−/−* peritoneal macrophages treated with Ac-LDL (120 μg/ml) in presence of the proteasome inhibitor MG132 (10 μM) for 12 h.

Source data are available online for this figure.

inhibition of p38 suppressed AKT activation in cultured macrophages and in atherosclerotic lesions analyzed from macrophage-specific p38α-deficient mice. Moreover, overexpression of constitutively active myristoylated AKT significantly attenuated the ER stress that

occurred with pharmacological inhibition of p38 in cultured macrophages. In our study, we observed that the absence of miR-21 markedly increases p38 phosphorylation but does not significantly influence AKT phosphorylation or PTEN expression. The later

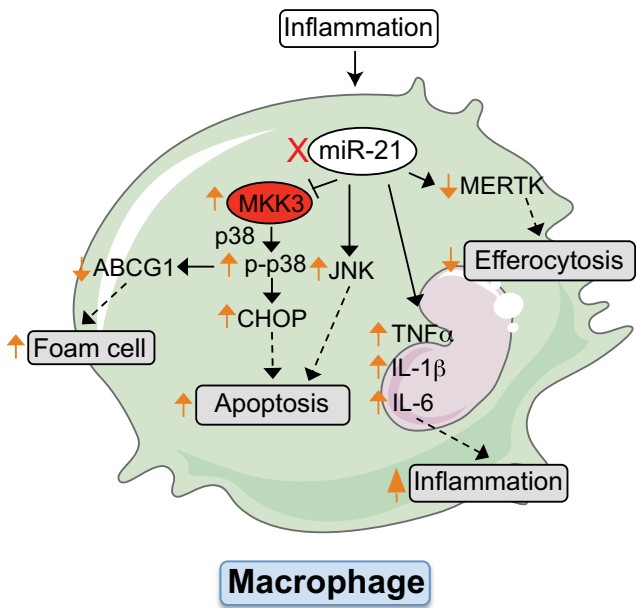

**Figure 8. Schematic diagram showing the role of miR-21 in macrophages during atherosclerosis.**

miR-21 expression influences foam cell formation, sensitivity to ER-stress-induced apoptosis, and phagocytic clearance capacity. Absence of miR-21 in macrophages is pro-inflammatory, increases the expression of the miR-21 target gene MKK3, promoting the induction of p38-CHOP and JNK signaling. Both pathways enhance macrophage apoptosis and promote the post-translational degradation of ABCG1, a transporter that regulates cholesterol efflux in macrophages.

observation was surprising because PTEN, a phosphatase that regulates AKT phosphorylation, is a well-known miR-21 target gene (Meng *et al*, 2007; Perdiguero *et al*, 2011).

Our work further demonstrates that the absence of miR-21 in macrophages significantly increased foam cell formation *in vitro*, which may also contribute to the increased apoptosis observed *in vivo*, as well as the overall increase in plaque formation. Interestingly, we found that miR-21-deficient macrophages have an impaired ability to efflux cholesterol to HDL, while efflux to ApoA1 is unaffected. Consistent with this, expression of the HDL-dependent cholesterol transporter ABCG1 was reduced in macrophages lacking miR-21, while expression of ABCA1, which primarily mediates efflux of cholesterol to poorly lipidated ApoA1, was similar to WT cells. ABCG1 expression is regulated at post-transcriptional level by JNK and p38. To this regard, Hedrick *et al* demonstrated that activation of p38 and JNK pathways by treating macrophages with eicosanoids inhibited ABCG1 and attenuated cholesterol efflux (Nagelin *et al*, 2009). By contrast, pharmacological inhibition of p38 and JNK signaling pathways impairs ABCG1 phosphorylation and degradation. Interestingly, we found that inhibition of p38 in miR-21-deficient macrophages increase ABCG1 protein stability and expression. Together, these findings indicate that the increased apoptosis observed in atherosclerotic plaques of mice transplanted with miR-21-deficient macrophages might be mediated by a combined effect of increased lipid loading and apoptosis susceptibility.

As introduced above, in addition to macrophages apoptosis, the ability of phagocytes to clear apoptotic cells contributes to lesional

necrosis, an important feature of advanced and unstable plaques (Tabas, 2010; Tabas *et al*, 2015). Interestingly, we found that the absence of miR-21 in macrophages results in a significant reduction in apoptotic cell uptake. This finding correlates with a significant reduction in MERTK expression, a phagocyte receptor implicated in apoptotic cell recognition. We next explored the molecular mechanism by which miR-21 controls MERTK levels and found that miR-21 does not influence *Mertk* mRNA levels or protein cleavage in response to LPS. Further experiments are needed to identify how miR-21 controls the expression of MERTK at a post-transcriptional level. These results support a model in which the absence of miR-21 increases macrophage apoptosis and impairs efficient phagocytosis of apoptotic macrophages, leading to increased plaque necrosis and accelerated atherosclerosis (Fig 8).

In summary, the data herein shed light on the important role of macrophage miR-21 during the progression of atherosclerosis. In this complex scenario, we demonstrate that absence of miR-21 controls macrophage foam cell formation, apoptosis, efferocytosis, and the inflammatory response associated to the resolution of inflammation. These effects in turn account for the adverse plaque remodeling observed in mice lacking miR-21 in hematopoietic cells (Fig 8). While this study provides definitive evidence of how miR-21 in hematopoietic cells impacts the progression of atherosclerosis and elucidates the primary mechanisms by which this occurs, further experiments are still needed to more fully define the complex network of genes controlled by miR-21, which may also be involved in mediating these cellular processes.

# Materials and Methods

### Animals and bone marrow transplantation

Male C57BL/6 (WT), *Ldlr*−/− (Ldlrtm1Her/J, Jax# 002207), or *miR-21*−/− (Mir21atm1Yoli/J, Jax # 016856) mice were purchased from The Jackson Laboratory (Bar Harbor, ME, USA) and kept under constant temperature and humidity in a 12-h controlled dark/light cycle. Eight-week-old male *Ldlr*−/− mice were lethally irradiated twice with a dose of 550 rads (5.5 Gy) using a cesium source 4 h before transplantation (Rotllan *et al*, 2015). Bone marrow was collected from femurs of WT or *miR-21*−/− mice by flushing with sterile Opti-MEM. Each recipient mouse was injected with $2 \times 10^6$ bone marrow cells through retro-orbital injection. Four weeks after bone marrow transplantation (BMT), peripheral blood was collected by retro-orbital venous plexus puncture for PCR analysis of bone marrow reconstitution.

Accelerated atherosclerosis was induced by feeding mice for 12 weeks with a Western-type diet (WD) containing high-cholesterol diet (1.25% cholesterol; D12108; Research Diets, Incorporated, New Brunswick, NJ, USA; Ulrich *et al*, 2016). All of the experiments were approved by the Institutional Animal Care Use Committee of Yale University School of Medicine.

### Lipoprotein profile and lipid measurements

Mice were fasted for 12–14 h before blood samples were collected by retro-orbital venous plexus puncture. Then, plasma was separated by centrifugation and stored at −80°C. Total plasma

cholesterol and triglycerides were enzymatically measured (Wako Pure Chemicals Tokyo, Japan) according to the manufacturer's instructions. The lipid distribution in plasma lipoprotein fractions was assessed by fast-performed liquid chromatography gel filtration with 2 Superose 6 HR 10/30 columns (Pharmacia Biotech, Uppsala, Sweden; Goedeke et al, 2014).

## Histology, immunohistochemistry, and morphometric analyses

After 12 weeks of WD feeding, mice were anesthetized and euthanized. Mouse hearts were perfused with 10 ml of phosphate-buffered saline (PBS; Invitrogen) at harvest and then submersed 4% paraformaldehyde (PFA) overnight. Then, hearts were put in 30% sucrose until the next day. Finally, hearts were embedded in OCT and frozen. Serial sections were cut at 6 μm thickness using a cryostat, and several morphometric and immunohistochemistry analyses were performed as previously described (Ulrich et al, 2016). Every third slide from the serial sections was stained with hematoxylin and eosin (H&E), and each consecutive slide was stained with Oil Red O for quantification of the lesion area. Aortic lesion size of each animal was obtained by averaging the lesion areas in three sections from the same mouse. The fibrous cap and necrotic core area were measured as a percentage of the total plaque area from the three sections from the same mouse. Necrotic core was defined as a clear area that was H&E free (Seimon et al, 2009). Boundary lines were drawn around these regions, and the area measurements were obtained by image analysis software (see below). Fibrous cap thickness was quantified by choosing the largest necrotic core from triplicate sections and taking a measurement from the thinnest part of the cap, determined by measuring the area between the outer edge of the cap and the necrotic core boundary (Seimon et al, 2009). Collagen content was assessed by Picrosirius red staining of consecutive slides from serial sections. CD68 staining was used as a macrophage marker using consecutive slides from three serial sections. Apoptotic cells in lesions were detected by TUNEL after proteinase K treatment, using the In Situ Cell Death Detection kit, TMR red (Roche, Basel, Switzerland) according to the manufacturer's instructions. Nuclei were counterstained with DAPI for 10 min. The data are expressed as the number of TUNEL-positive cells per millimeter squared cellular lesion area. Proliferation of cells in each lesion was detected by Ki67 staining (1:100; Abcam, Cambridge, MA, USA). Percentage of proliferating cells were calculated as the number of positive Ki67-labeled nuclei divided by the number of DAPI-stained nuclei. NIH ImageJ software (National Institutes of Health, Bethesda, MD, USA) was used for all the quantifications.

## En face Oil Red O staining

After incubation in 4% PFA overnight, the adventitia was thoroughly cleaned under a dissecting microscope, and the aorta was cut open longitudinally and pinned on to a silicone plate. So that the lesion area could be calculated, aortas were stained with Oil Red O (Sigma) before the analysis. Aortas opened up longitudinally were briefly rinsed with 78% methanol, stained with 0.16% Oil Red O solution for 50 min and then destained in 78% methanol for 5 min, and mounted on microscopic slides with aqueous mounting medium

(Stephens Scientific). The lesion area was quantified as percent of Oil Red O staining area in total aorta area (Fernandez-Hernando et al, 2007).

## Detection of miR-21 by in situ hybridization

To perform the in situ hybridization, we followed the protocol of Silahtaroglu (2014) with some modifications. Briefly, 6-μm cryosections from adult mouse heart were fixed in 4% paraformaldehyde and acetylated in acetic anhydride/triethanolamine, followed by two washes in PBS. Sections were then pre-hybridized in hybridization solution (50% formamide, 5× SSC, 0.5 mg/ml yeast tRNA, 1× Denhardt's solution) at 25°C below the predicted Tm value of the LNA probe for 30 min. DIG-labeled probes (3 pmol; LNA miRCURY probe; Exiqon) were hybridized to the sections for 2.5 h at the same temperature as pre-hybridization. After hybridization, we washed three times in 0.1× SSC at 60°C, and the in situ hybridization signals were detected using the tyramide signal amplification system (Perkin-Elmer) according to the manufacturer's instructions. Slides were mounted in Prolong Gold containing DAPI (Invitrogen) and analyzed with a Zeiss Axiovert 2000M fluorescence microscope (Carl Zeiss).

## Cell culture

Peritoneal macrophages from adult male WT or *miR-21*$^{-/-}$ mice were harvested by peritoneal lavage 4 days after intraperitoneal injection of thioglycollate (3% w/v). Cells were plated in RPMI 1640 medium supplemented with 10% fetal bovine serum, 100 U/ml penicillin, and 100 U/ml streptomycin. After 4 h, non-adherent cells were washed out, and macrophages were incubated in fresh medium containing DMEM, 20% fetal bovine serum, and 20% L-cell conditioned medium for 2 days, and cells were maintained in culture as an adherent monolayer adding fresh medium every day (Aryal, Rotllan et al, 2016). Peritoneal macrophages were used to determine the effect on cell survival, efferocytosis, and signaling pathways. To induce ER stress or cell death, cells were treated with tunicamycin (5 μg/ml), thapsigargin (2 μM), or loaded with free cholesterol via incubation with Ac-LDL (120 μg cholesterol/ml) + ACAT inhibitor [Sandoz 58035 (10 μg/ml)]. In some instances, activation of p38 was inhibited by pretreatment for 2 h with SB203580 (10 μM). For cholesterol loading, cells were incubated with Ac-LDL (120 μg cholesterol/ml). More detailed descriptions can be found in the figure legend describing these experiments.

Bone marrow-derived macrophages (BMDMs) from adult male WT or *miR-21*$^{-/-}$ mice were harvested and cultured in Iscove's modified Dulbecco's medium (IMDM) supplemented with 20% fetal bovine serum and 20% L-cell conditioned medium. After 7 days in culture, non-adherent cells were eliminated, and adherent cells were harvested for the assays (Aryal et al, 2016). BMDMs from WT and *miR-21*$^{-/-}$ mice were used to determine the effect on inflammatory cytokines. Briefly, BMDMs were stimulated for 6 h with LPS (10 ng/ml; Sigma). At the end of the treatment, cells were extensively washed with PBS and RNA and protein were isolated for analysis of mRNA and protein levels as described below.

## RNA isolation and quantitative real-time PCR

Total RNA from mouse aortas was isolated using the Bullet Blender Homogenizer (Next Advance, Averill Park, NY, USA) in TRIzol reagent (Invitrogen) according to the manufacturer's protocol. Total RNA from BMDMs was isolated using TRIzol reagent. One microgram of total RNA was reverse-transcribed using the iScript RT Supermix (Bio-Rad, Hercules, CA, USA), following the manufacturer's protocol. Quantitative real-time PCR was performed in triplicate using iQ SYBR green Supermix (Bio-Rad) on a Real-Time Detection System (Bio-Rad). The mRNA level was normalized to ribosomal RNA 18S as a housekeeping gene. The following mouse primer sequences were used: CD68, 5′-CCAATTCAGGGTGGAAGAAA-3′ and 5′-CTCGGGC TCTGATGTAGGTC-3′; IL-6, 5′-AGTTGCCTTCTTGGGACTGA-3′ and 5′-TCCACGATTTCCCAGAGAAC-3′; TNF-α, 5′-CCCTCACACTCAGATC ATCTTCT-3′ and 5′-GCTACGACGTGGGCTACAG-3′; IL-1β, 5′-CCAAA ATACCTGTGGCCTTGG-3′ and 5′-GCTTGTGCTCTGCTTGTGAG-3′; MerTK, 5′-TGCGTTTAATCACACCATTGGA-3′ and 5′-TGCCCCGAGC AATTCTTTC-3′; 18S, 5′-TTCCGATAACGAACGAGACTCT-3′ and 5′-T GGCTGAACGCCACTTGTC-3′.

Mature miR-21 levels were detected using TaqMan miRNA Assay kit (Life Technologies) according to the manufacturer's protocol. Quantitative real-time PCR was performed using TaqMan Universal Master Mix (Life Technologies; Chamorro-Jorganes et al, 2011, 2014; Suarez et al, 2007, 2008, 2010). RNA U6 was used for normalization.

## RNA sequencing

Total RNA from unstimulated BMDMs was extracted as described above, and RNA was purified using and miRNA isolation Kit (Qiagen) followed by DNase treatment to remove genomic contamination using RNA MinElute Cleanup (Qiagen). The purity and integrity of total RNA samples were verified using the Agilent Bioanalyzer (Agilent Technologies, Santa Clara, CA, USA). rRNA was depleted from RNA samples using Ribo-Zero rRNA Removal Kit (Illumina). Small RNA libraries from WT BMDMs were performed using TrueSeq Small RNA Library preparation (Illumina) and were sequenced for 45 cycles on Illumina HiSeq 2000 platform (2 × 50 bp read length). MicroRNA sequencing results were trimmed and mapped to miRBase mouse stem-loop sequences (http://www.mir base.org/) using the Bowtie alignment (http://bowtie-bio.sourcef orge.net/index.shtml) program (version 0.12.7). The alignment reads were normalized as proportion of total reads mapped to any know miRNAs in each sample. The data discussed in this publication have been deposited in NCBI Gene Expression Omnibus and are accessible through GEO Series accession number GSE97622 (https://www.ncbi.nlm.nih.gov/geo/query/acc.cgi?acc = GSE97622).

## Western blotting

Cells were lysed in ice-cold buffer containing 50 mM Tris–HCl, pH 7.5, 125 mM NaCl, 1% Nonidet P-40, 5.3 mM NaF, 1.5 mM NaP, 1 mM orthovanadate, 1 mg/ml protease inhibitor cocktail (Roche), and 0.25 mg/ml Pefabloc, 4-(2-aminoethyl)-benzenesulfonyl fluo-ride hydrochloride (AEBSF; Roche). Cell lysates were rotated at 4°C for 1 h before the insoluble material was removed by centrifugation at 12,000 × g for 10 min. After normalizing for equal protein concentration, cell lysates were resuspended in SDS sample buffer before separation by SDS–PAGE. To induce and harvest sMER, cultures were typically from 12-well tissue culture-treated plates that were overlaid with 50 ng/ml LPS in serum-free medium for the indi-cated times. Cell supernatants were concentrated 10-fold with Amicon Ultra centrifugal filters (10,000 molecular weight cut-off; Thorp et al, 2011).

Western blots were performed using the following antibodies: ABCG1 (#NB400-132; 1:1,000), COX-2 (#160106; 1:1,000), and iNOS (#2982; 1:1,000) were obtained from Novus, Cayman, and Cell Signaling Technology, respectively; a mouse monoclonal anti-body against ABCA1 (#ab18180, 1:1,000) was obtained from Abcam; a mouse monoclonal antibody against HSP-90 (#610419, 1:1,000) from BD Biosciences. AKT (pan; #2920; 1:1,000), phos-pho-Akt (thr308; #13038; 1:1,000), CHOP (#2895; 1:1,000), PTEN (#9188; 1:1,000), MKK3 (#8535; 1:1,000), Phospho-MKK3 (Ser189)/MKK6 (Ser207) (#12280; 1:1,000), Phospho-SAPK/JNK (Thr183/Tyr185) (#4668; 1:1,000), SAPK/JNK (#9252; 1:1,000), PDCD4 (#9535; 1:1,000), Phospho-p38 MAPK (Thr180/Tyr182; #9211; 1:1,000) were from Cell Signaling Technology. Anti-p38α (612168; 1:1,000) was purchased from BD Biosciences and the MKP-1 (SC-1199; 1:1,000) from Santa Cruz Biotechnology. Anti-mouse Mer primary antibody (AF591; R&D systems). Secondary antibodies were fluorescence-labeled antibodies obtained from LI-COR Biotechnology. Protein bands were visualized using the Odyssey Infrared Imaging System (LI-COR Biosciences, Lincoln, NE, USA). Densitometry analysis of the gels was carried out using NIH ImageJ software (http://rsbweb.nih.gov/ij/). For quan-tification of Western blot analysis, levels of protein of interest are expressed in arbitrary units after correction by loading control. For phosphorylated proteins, levels are corrected by respective total protein levels.

## miRNA mimic transfections

Mouse peritoneal macrophages were transfected with 40 nM of miR-21 mimic (Dharmacon) utilizing RNAimax (Invitrogen) following manufacturer standard protocols. All experimental control samples were treated with an equal concentration of a non-targeting control mimic sequence (CM, Dharmacon) for use as controls for non-sequence-specific effects in miRNA experi-ments (Chamorro-Jorganes et al, 2016). Verification of miR-21 overexpression was determined using qRT-PCR as described above.

## Flow cytometry analysis of blood and aorta leukocyte, apoptosis, and Dil-Ox-LDL uptake

The following antibodies were used for flow cytometric analyses: anti-CD4-APCcy7, RM4-5 (BioLegend); anti-CD8-APCcy7, 53-6.7 (BioLegend); anti-B220-APCcy7, RA3-6B2 (BioLegend); anti-CD127-APCcy7 (IL-7Pα), A7P34 (BioLegend); anti-CD90.2-APCcy7, 30-H12 (BioLegend); anti-NK1.1-APCcy7, PK136 (BioLegend); anti-Ly-6G-APCcy7, 1A8 (BioLegend); anti-TER119-APCcy7, TER119 (BioLe-gend); anti-CD11b-PB, M1/70 (BioLegend); anti-CD11c-Percp Cy5.5, HL3 (BioLegend); anti-F480-APC, BM8 (BioLegend); anti-CD45.2-FITC, 104 (eBiosciences); anti-CD3–PE, 17A2 (BioLegend); anti-CD115, AFS98 (eBiosciences). We used a 1:200 dilution for all anti-bodies. All antibodies were titrated to determine optimal signal-to-

    

noise separation and minimize background fluorescence. Gating strategy was set up using Fluorescence Minus One (FMO) control.

Blood was collected by retro-orbital puncture in heparinized microhematocrit capillary tubes. Erythrocytes were lysed with ACK lysis buffer (155 mM ammonium chloride, 10 mM potassium bicarbonate, and 0.01 mM EDTA, pH 7.4). White blood cells were resuspended in 3% fetal bovine serum in PBS, blocked with 2 mg/ml FcgRII/III, then stained with a cocktail of antibodies. Monocytes were identified as $CD115^{high}$ and subsets as $Ly6-C^{high}$ and $Ly6-C^{low}$. Neutrophils were identified as $CD115^{low}$, $Ly5-C^{high}$, and $Ly6-G^{high}$ (Aryal *et al*, 2016; Ulrich *et al*, 2016). For sorting cells from peripheral blood, monocytes were identified as $CD45^{+}$, $CD115^{high}$, and $CD11b^{+}$, whereas neutrophils were identified as $CD45^{+}$, $CD115^{low}$, and $CD11b^{+}$. T lymphocytes were identified as $CD45^{+}$, $CD3^{+}$ cells. Sorted cells were resuspended in TRIzol for RNA isolation and determination of miR-21 expression as described above.

For aortic tissue, the entire aorta was digested (from the root to the iliac bifurcation) according to a method previously published (Butcher *et al*, 2011). The procedure involves the perfusion of the aorta (20 ml PBS) prior to digestion. Aortic tissue was cut in small pieces and subjected to enzymatic digestion with 400 U/ml collagenase I, 125 U/ml collagenase XI, 60 U/ml DNase I, and 60 U/ml hyaluronidase (Sigma-Aldrich) for 1 h at 37°C while shaking. Total viable cell numbers were obtained using Trypan Blue (Cellgro) prior flow cytometric analysis as described previously (Aryal *et al*, 2016). Monocytes were identified as $CD11b^{+}$, lineage$^{-/low}$ (lineage defined as $CD4^{+}$ $CD8^{+}$ $CD127^{+}$ $CD90.2^{+}$ $B220^{+}$ $NK1.1^{+}$ $Ly-6G^{+}$ $Ter119^{+}$), $CD11c^{-}$, F4/80 low, and $Ly-6C^{high}$. Macrophages were identified as $CD11b^{+}$, lineage$^{-/low}$, $CD11c^{-}$, and $F4/80^{high}$. Neutrophils were identified as $CD11b^{+}$, lineage positive, and $CD11c^{-}$. For sorting cells from mouse aorta, monocytes/macrophages were identified as $CD45^{+}$, $CD11b^{+}$, lineage$^{-}$, $CD11c^{-}$. Neutrophils were defined as $CD45^{+}$, $CD11b^{+}$, lineage$^{+}$, and $CD11c^{-}$. Lineage$^{+}$ cells were $CD45^{+}$ cells not including monocytes/macrophages and neutrophils ($Cd11b^{-}$). $CD45^{-}$ isolated cells were aortic non-leukocytic cells. Sorted cells were resuspended in TRIzol for RNA isolation and determination of miR-21 expression as described above.

Peritoneal macrophages from WT or $miR-21^{-/-}$ mice were analyzed for surface expression of MERTK by flow cytometry. After the indicated treatments, confluent monolayers were washed twice in Hanks balanced salt solution (HBSS, Gibco, Grand Island, NY, USA) and harvested. Cells were resuspended in 3% fetal bovine serum in PBS, blocked with 2 mg/ml FcgRII/III for 15 min at room temperature. Mertk surface expression was analyzed by using goat anti-mouse Mer primary antibody (AF591; R&D systems, dilution 1:100) and incubation for 45 min on ice. Cells were washed with PBS and incubated with anti-goat Alexa 488 secondary antibody for 30 min on ice. For a negative control, a separate set of cells were incubated without antibody under same conditions followed by the incubation with the secondary labeled antibody as indicated above. In both cases, immunostained cells were then washed twice with cold PBS and analyzed on a flow cytometer; 10,000 gated viable cells per sample were analyzed. The results are expressed in terms of specific geometric mean intensity of fluorescence (M.I.F.) after subtracting fluorescence of cells incubated with isotype control.

Apoptotic cells were analyzed with fluorescein isothiocyanate (FITC)-conjugated annexin V staining (1:20, BioLegend) together with propidium iodide (PI) dead cell counterstain according to the manufacturer's recommendations (Fernandez-Hernando *et al*, 2007). Percentage of non-viable cells are the sum of late apoptotic cells (Annexin $V^{+}/PI^{+}$) and the early apoptotic cells (Annexin $V^{+}/PI^{+}$).

DiI-Ox-LDL lipoproteins were oxidized and labeled with the fluorescent probe DiI (Molecular Probes, Invitrogen) as previously described (Suarez *et al*, 2004). For the uptake assays, peritoneal macrophages from WT or $miR-21^{-/-}$ mice were washed once in 2× PBS and incubated in fresh media containing DiI-Ox-LDL (30 μg cholesterol/ml) for 2 h at 37°C. Then, cells were washed and resuspended in 1 ml of PBS and analyzed by flow cytometry (Goedeke *et al*, 2015). The results are expressed in terms of M.I.F. after subtracting the autofluorescence of cells incubated in the absence of DiI-Ox-LDL.

All flow cytometry was performed using a BD LSRII (BD Biosciences). All flow cytometry data were analyzed using FlowJo software v8.8.6 (Tree Star, Inc.). In some instances, blood or aortic cells were sorted on a BD FACSAria (BD Biosciences).

### Efferocytosis assays

Peritoneal macrophages were recovered from mice as described above; $5 \times 10^5$ cells were plated on sterile glass coverslips with RPMI, 20% fetal bovine serum, and 20% L-cell conditioned medium for 2 days. To generate apoptotic cells (AC), Jurkat cells were labeled with CellTracker Red (Invitrogen) according to the manufacturer's instructions. Labeled cells were UV-irradiated for 6 min using 8W bulb, and after the irradiation, cells were incubated at 37°C for 2 h to induce apoptosis. Fluorescent AC were added to peritoneal macrophages at a 1:1 ratio (ACs:macrophages) and cultured at 37°C for 60 min in RPMI supplemented with 10% FBS. After incubation with ATs, macrophages were gently washed several times with cold PBS with EDTA 0.6 mM to remove free AC. Cells were then fixed with 4% paraformaldehyde, and efferocytosis engulfment was scored with a Zeiss Axiovert 2000M fluorescence microscope (Carl Zeiss). Efferocytosis engulfment was expressed as phagocytic index: number of cells ingested per total number of macrophages × 100. Macrophages were visualized with F4/80 antibody.

### Cholesterol efflux assays

Peritoneal macrophages from WT and $miR-21^{-/-}$ mice were cultured at a density of $1 \times 10^6$ cells per well 1 day prior to loading with 0.5 mCi/ml [$^3$H]-cholesterol for 24 h with or without T0901317 (3 μM) for 12 h (Rayner *et al*, 2010). Cells then were washed twice with PBS and incubated in RPMI 1640 medium supplemented with 2 mg/ml fatty acid-free bovine serum albumin (FAFA media) in the presence of an acetyl-coenzyme A acetyltransferase (ACAT) inhibitor (2 mM; Novartis Corporation, New York, NY, USA) for 4 h prior to the addition of 50 μg/ml human ApoA1 in FAFA or HDL (Intracell) media. Supernatants were collected after 6 h and expressed as a percentage of [$^3$H]-cholesterol in the media per total cell [$^3$H]-cholesterol content (total effluxed [$^3$H]-cholesterol + cell-associated [$^3$H]-cholesterol).

**The paper explained**

**Problem**
Atherosclerosis is the leading cause of death in Western societies. This immunemetabolic disease is characterized by the accumulation of lipids and chronic inflammation in the artery wall. Recent studies have demonstrated the critical importance of non-coding RNAs in regulating macrophage lipid homeostasis and inflammation during the progression of atherosclerosis. While miR-21 expression has been associated with conditions involving by impaired immune response including psoriasis, chronic bacterial inflammation, and asthma, its role during atherogenesis remains unknown. Therefore, we hypothesized that the expression of miR-21 in macrophages might contribute to atherosclerosis development.

**Results**
We show that absence of miR-21 in hematopoietic cells enhances atherogenesis and promotes adverse plaque remodeling, characterized by increased plaque necrosis and fibrous cap thinning. We found that miR-21 deficiency in macrophages enhances apoptosis and attenuates their phagocytic capacity. Moreover, we demonstrate that absence of miR-21 reduces ABCG1 expression and cholesterol efflux, thus promoting the lipid accumulation in macrophages. Mechanistically, we demonstrate that lack of miR-21 in macrophages increases the activation of MKK3 (miR-21 target gene) signaling pathways triggering apoptosis in response to ER stress.

**Impact**
Our findings define a major role for hematopoietic miR-21 during the progression of atherosclerosis and underscore the significance of miR-21 in regulating macrophage apoptosis, efferocytosis, and lipid metabolism during atherogenesis. Our findings suggest that therapies aimed to increase miR-21 levels in advanced atherosclerotic plaques might significantly reduce plaque necrosis and inflammation.

**Cellular cholesterol measurement and foam cell formation analysis**

Peritoneal macrophages from WT and *miR-21*$^{-/-}$ mice were plated on 12-well plates and incubated with or without acetylated LDL (Ac-LDL; 120 µg/ml). After 24 h, intracellular total cholesterol, cholesterol esters, and free cholesterol content were measured using the Amplex Red Cholesterol Assay Kit (Molecular Probes; Invitrogen), according to the manufacturer's instructions. For the foam cell formation assay, peritoneal macrophages from adult WT and *miR-21*$^{-/-}$ mice were exposed to 120 µg/ml ac-LDL for 24 h in the presence of 10% lipoprotein-deficient serum (LPDS), fixed by 4% PFA in PBS for 1 h and stained for 30 min with 0.36% Oil Red O (ORO) solution in 60% isopropanol. The mean area and integrated optical density (IOD) of Oil Red O-stained region per cell were quantified with 200 representative cells using the ImageJ software from the NIH.

**Statistical analysis**

Animal sample size for each study was chosen based on literature documentation of similar well-characterized experiments. The number of animals used in each study is listed in the figure legends. *In vitro* experiments were routinely repeated at least three times unless otherwise noted. No inclusion or exclusion criteria were used, and studies were not blinded to investigators or formally randomized. Data are expressed as average ± SD or ± SEM. Statistical differences were measured using an unpaired two-sided Student's *t*-test, one-way ANOVA with Bonferroni correction for multiple comparisons. Normality was checked using the Kolmogorov–Smirnov test. A nonparametric test (Mann–Whitney) was used when data did not pass the normality test. A value of $P \leq 0.05$ was considered statistically significant. Data analysis was performed using GraphPad Prism Software Version 7 (GraphPad, San Diego, CA, USA).

**Expanded View** for this article is available online.

## Acknowledgements

We thank Dr. Nathan Price (Yale University) for editing the manuscript, Dr. Zongzhi Liu (Yale University), Dr. Jenny Z. Xiang, and Dr. Olivier Elemento (Weill Cornell Medical College Genome Facility) for RNA sequencing analysis, Dr. Anton M. Bennett (Yale University) for critical suggestions and providing reagents, Dr. Zuzana Tobiasova (Yale University) for providing help with cell-sorting experiments. This work was supported by grants from the National Institutes of Health (R01HL107953 and R01HL106063 to CF-H; R01HL105945 and DRC P30 DK045735 to YS), the American Heart Association (16GRNT26420047 to YS; 16EIA27550005 to CF-H and 17SDG33110002 to NR), the American Diabetes Association (1-16-PMF-002 to AC-D), the Howard Hughes Medical Institute International Student Research Fellowship (to EA), the Foundation Leducq Transatlantic Network of Excellence in Cardiovascular Research MIRVAD (to CF-H), and the Ministerio de Economía y Competitividad e Innovación, Spain (SAF2015-70747-R to RB). CIBEROBN is an initiative of ISCIII, Spain.

## Author contributions

AC-D, CF-H, and YS conceived and designed the study and wrote the manuscript. AC-D, NR, XZ, MF-F, CR-H, EA, and LD performed experiments and analyzed data. RB provided reagents (Ox-DiI-LDL and Ac-LDL).

## Conflict of interest

The authors declare that they have no conflict of interest.

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
