## [Review Process File · EMBO Molecular Medicine]

Macrophage deficiency of miR-21 promotes apoptosis, plaque necrosis and vascular inflammation during atherogenesis

Alberto Canfrán-Duque, Noemi Rotllan, Xinbo Zhang, Marta Fernández-Fuertes, Cristina Ramírez-Hidalgo, Elisa Araldi, Lidia Daimiel, Rebeca Busto, Carlos Fernández-Hernando and Yajaira Suárez

*Corresponding authors: Carlos Fernández-Hernando and Yajaira Suárez
Yale University School of Medicine*

Review timeline:

Submission date:	20 December 2016
Editorial Decision:	06 February 2017
Revision received:	04 April 2017
Editorial Decision:	17 May 2017
Revision received:	30 May 2017
Accepted:	06 June 2017

Transaction Report:

Editor: Céline Carret

1st Editorial Decision

06 February 2017

Thank you for the submission of your manuscript to EMBO Molecular Medicine. We have now heard back from the three referees whom we asked to evaluate your manuscript.

You will see that all three referees are rather supportive of the study however, all three highlight technical issues including Western blots that should be submitted in full and statistical testing that should be reinforced by using alternative tests. It is our opinion that all suggested experiments and text modifications are reasonable and would improve the impact of the paper and I would therefore encourage you to address these in a major revision of your work. Please note that it is EMBO Molecular Medicine policy to allow only a single round of revision and that, as acceptance or rejection of the manuscript will depend on another round of review, your responses should be as complete as possible.

I look forward to receiving your revised manuscript.

***** Reviewer's comments *****

Referee #1 (Remarks):

This paper describes a role for miR-21 in atherosclerosis, attributing it to plaque macrophages. The data show that absence of miR-21 in hematopoietic cells (after bone marrow transfer) accelerates the progression of atherosclerosis, increases monocyte recruitment, and promotes macrophage apoptosis, resulting in larger necrotic cores and thinner fibrous caps.

The data are comprehensive as they touch on major mechanisms involved in atherosclerosis and macrophage biology. The manuscript is novel and the topic is interesting and timely.

It would be even better if data on a cell specific deletion were available. This is a limitation that should be discussed. As is, the observed changes in the atherosclerosis phenotype could also be caused by other marrow derived cells, e.g. neutrophils. What is the expression of miR-21 in those?

The expression of miR-21 in plaque macrophages was not formally shown. This could be done by sorting macrophages from aortas, followed by PCR.

Referee #2 (Remarks):

This manuscript by Canfrán-Duque is well-written and outlines interesting aspects of miR-21 in regulating a dynamic role between the balance of pro- and anti-inflammatory immune responses. The presentation of the manuscript is clear and concise although a slight rearrangement I feel would improve the overall structure. Each section is well-developed and the authors have addressed the initial query set out by the title. The conclusions are well supported by the comprehensive experimental results. The authors have contextualized their work in this ongoing area of research this by comparing similar findings to their results. However there are some issues that need to be addressed and clarified.

Major points:

1. Statistics seem to only consist of Student t-tests. Student t-test assumes normal distributions. Generally for in vivo models normal distributions should not be assumed? Some graphs have more than two groups but all comparisons are made using t-tests. Please revise and ensure correct analyses are used throughout the manuscript.
2. Statistical comparisons are said to compare mean +/- standard error of the mean (SEM). SEMs must be a minimum n=3 in triplicate. However, this is not the case for all statistical analysis in the manuscript.
3. It would be clearer if all flow cytometry data expressed mean intensity fluorescence (MIF) or % populations not both. Furthermore, in relation to the flow cytometry data please note the following:
 - a. Gating schemes should always be shown and if not shown in text need to be shown in supplemental data
 - b. Methods used to gate - based on unstained samples, fluorescence minus one controls, isotype controls etc.
 - c. How compensation was carried out? Multi-color flow cytometry usually uses stained beads
 - d. If using MFI - compare means/medians, its important to show that the flow cytometer machine has been calibrated on a regular basis otherwise MFIs can change over time.
 - e. Example of flow data in graphs presented shown in plots in supplemental data, if not in body of text.
4. Please include protein sizes in all Western blots.
5. In figure 1C an appropriate control must be included, for example, Ldlr^{-/-}miR-21^{-/-} with no bone marrow transplant. The control shown in the supplementary data for this is from a Ldlr^{-/-}Cav-1^{-/-}

mouse which is not ideal.

6. Figure 5- there is no explanation for MERTK hi and lo. Please include a description. More importantly, the Western blot shown in Figure 5 A does not support the conclusion that total MERTK levels were significantly reduced in MiR21 deficient macrophages. In fact the figure shown in Fig 5A to me looks like there is no change. Please replace with a blot that clearly shows the decrease in expression. Same comment for Figure 5F- the increase in iNOS is not evident in the blot shown.

7. Housekeeping genes used for qRT-PCR are not always stated.

Minor points:

1. miR-21 is a key signal mediating the balance of pro- and anti-inflammatory responses this should be stated early on, not towards the end of the introduction after discussing it in both cases.

2. qRT-PCR changes from fold-change to relative expression - please ensure consistency throughout

3. p-values not shown on each graph and not stated in figure legends. Only general $p < 0.05$ considered significant mentioned.

4. Some graphs show p-value, stars and ns - please ensure consistency in the way the data is presented.

5. Figure 7 should have a figure legend describing the image - every figure should be stand alone.

6. Italics for in vivo, in vitro etc.

7. Please use the correct nomenclature of cytokines i.e TNF- α , IL-6, IL-1 β , NF- κ B etc. not TNF α , IL6, IL1 β , NF-KB

8. Please correct typographical errors for example Sheedy et. al. not Sheddy et. al.

Referee #3 (Remarks):

The manuscript reports that loss of miR-21 in hematopoietic cells accelerates the progression of atherosclerosis and is associated with increased apoptosis and necrosis and a reduced fibrous cap. These findings add to the current understanding of both atherosclerotic progression and miR-21 function. However I think the manuscript could be improved by addressing the following comments.

1.) Figure 1c shows miR-21 in situ hybridization (green staining) in plaques of Ldlr-/-miR-21-/- mice transplanted with WT bone marrow but Fig. S1 also shows significant green staining in what are described as Ldlr-/-miR-21-/- mice that were not transplanted. While the plaque area in Fig S1 may be devoid of staining there is significant staining elsewhere in the image despite the fact that the mice lack miR-21. I assume the mice are miR-21 null and not a conditional strain as that is all that appears on Jackson Labs website at this time but I did not see that actually noted in the manuscript. This should be clarified.

2.) The figure legend for 1d makes no mention of the included Oil Red O images or the associated quantification.

3.) The lesion area in the cross sectional images in Fig. 1d is greater in the miR-21-/- transplanted mice although there is apparently no significant difference in the oil red o staining within the cross sectional lesion area. However, in Figure 1 E the 3 images labeled as WT->Ldlr-/- appear to have more oil red o staining in the aortic and thoracic arch regions while the associated quantification shows the opposite i.e. less oil red o in the WT transplanted mice. I expect the labels in 1 E may be switched.

- 4.) Figure 2A. Was the zoomed in section (right panel) or the whole image (left panel) used to quantify the necrotic core and fibrous cap regions? If the zoomed in section, why was this specific region of the plaque chosen to quantify?
- 5.) The figure legend for Fig 2 states "all data represent mean \pm SEM (n=9-11 per group)" yet the TUNEL+ graph in 2b shows on 7 WT and 6 mir-21 data points.
- 6.) The y axis in 3A has a break between 4 and 5. Why? I assume the Y axis of the plots in 1C should be "Lineage PE" (not lineage PE) although I didn't see that that antibody cocktail mentioned in the methods.
- 7.) The sheer number of bands in figure 4 makes it difficult to identify the important differences between conditions/ strains. I do not see value in showing the primary western data for so many proteins that do not appear to change across the conditions. There is also no quantification of the differences shown or referenced in the text.
- 8.) No quantification of the MERTK levels shown in 5A is given but the text describes it as significantly reduced in miR-21 KO cells. The images in my copy do not look significantly different.
- 9.) 5b - What does the # in the panel represent? It is not referenced in the figure legend.
- 10.) 5C why is COX2 shown here? How does that data compare to the COX2 data shown in 5F with respect to differences observed between WT and KO cells?
- 11.) The differences between WT and miR-21 $^{-/-}$ in Figure 6 appear to be relatively small across all the panels but the nearly 2x difference in ABCG1 levels at 24 hrs shown in 6E is particularly hard to appreciate. I think the authors should critically evaluate the necessity and/or alternative presentation of this data.

1st Revision - authors' response

04 April 2017

Referee #1:

1- It would be even better if data on a cell specific deletion were available. This is a limitation that should be discussed. As is, the observed changes in the atherosclerosis phenotype could also be caused by other marrow-derived cells, e.g. neutrophils. What is the expression of miR-21 in those?

Bone marrow transplantation (BMT) experiments using gene-targeted mice provide a useful approach to examine the contribution of macrophage gene expression to atherogenesis *in vivo*. However, BMT experiments, as pointed by the Reviewer, cannot rule out the potential effect of miR-21 expression in other hematopoietic cells besides macrophages (e.g. neutrophils) during the progression of atherosclerosis (Weber, Zernecke et al., 2008). An alternate approach would be the use of double mutant mice harboring both the LysMcre allele and the loxP-flanked miR-21 gene. LysMcre mice allow for both fairly specific and highly efficient Cre-mediated deletion of loxP-flanked target genes in myeloid cells (83-98% in mature macrophages (Cross, Mangelsdorf et al., 1988), while monocytes are not efficiently targeted, and near 100% in granulocytes, which include neutrophils) (Clausen, Burkhardt et al., 1999). However, partial deletion (16%) could be detected in CD11c+ splenic dendritic cells, which are closely related to the monocyte/macrophage lineage (Albarran-Juarez, Kaur et al., 2016). Cumulatively, this approach would not avoid the effect of the absence of miR-21 in other cells types and would also not efficiently remove miR-21 from circulating monocytes. Although the arterial wall contains a large number of resident macrophages and some resident dendritic cells, atherosclerosis drives a rapid influx of inflammatory monocytes and neutrophils (Weber et al., 2008). Both methodologies have their limitations; with the BMT the main issue is the hematopoietic cells will be deficient of miR-21; with the conditional deletion the cell types that would have miR-21 ablation are reduced, but inability to target monocytes is also a concern given the importance of recruited monocytes during atherogenesis. As indicated by the Reviewer we provide a brief discussion related to that matter within the manuscript.

We are sensitive to the Reviewer concern regarding the expression of miR-21 in other hematopoietic cells including neutrophils since their role in atherosclerosis has also been demonstrated. miRNA expression in hematopoietic lineages reveals that among other miRNAs, miR-21 was specific to macrophages and T cells. In line with this, a recent study performed miRNA profiling in blood cells revealed different patterns and different expression levels of miRNA in platelets, B cells, T cells, granulocytes or erythrocytes, unfortunately monocytes were not included in the analysis. In this study, miR-21 was only among the 10 most abundant miRNAs in T cells where it was the 5th most abundant but just accounting for 4% of the total mass of miRNAs. Despite this, we have analyzed the expression of miR-21 in different hematopoietic cell populations, such as, T cells, neutrophils and monocytes. Our data show that circulating neutrophils express higher levels of miR-21 than in T cells. Regardless of this miR-21 was highly expressed in monocytes, which is in agreement with previous reports. These results have been included as **Fig S1** and briefly described within the manuscript.

2- The expression of miR-21 in plaque macrophages was not formally shown. This could be done by sorting macrophages from aortas, followed by PCR.

This is a very important point raised by the reviewer. Thus, we sorted different cell types from aortas of *Ldlr*^{-/-} fed a WD for 12 weeks and, confirmed that the expression of miR-21 in the monocyte/macrophage compartment was higher compared to the rest of cellular components, including neutrophils, sorted from the plaque. These results are shown in the new (**Fig 1 C**) and accordingly described within the manuscript.

Referee #2:

We would like to thank the Reviewer for finding that our manuscript “*is well-written... is clear and concise*” and noting that our “conclusions are well supported by the comprehensive experimental results.” Your comments and suggested experiments have unequivocally provided strength to our manuscript. Your remarks are italicized and followed by our answers.

Major points:

1- *Statistics seem to only consist of Student t-tests. Student t-test assumes normal distributions. Generally for in vivo models normal distributions should not be assumed? Some graphs have more than two groups but all comparisons are made using t-tests. Please revise and ensure correct analysis are used throughout the manuscript.*

We deeply apologize for the poor description of the statistical analysis performed. As the reviewer pointed in some instances the figure legends incorrectly state the use of Student t-test. We have revised all the statistical analysis performed and modified accordingly where required. Briefly, animal sample size for each study was chosen based on literature documentation of similar well-characterized experiments. The number of animals used in each study is listed in the figure legends. *In vitro* experiments were routinely repeated at least three times unless otherwise noted. Data are expressed as average \pm SD or \pm SEM as indicated. Statistical differences were measured using an unpaired two-sided Student's *t*-test, and/or one-way ANOVA with Bonferroni correction for multiple comparisons. Normality was checked using the Kolmogorov-Smirnov test. A nonparametric test (Mann-Whitney) was used when data did not pass the normality test. A value of $p \leq 0.05$ was considered statistically significant. Data analysis was performed using GraphPad Prism Software Version 7 (GraphPad, San Diego, CA).

2- *Statistical comparisons are said to compare mean +/- standard error of the mean (SEM). SEMs must be a minimum n=3 in triplicate. However, this is not the case for all statistical analysis in the manuscript.*

As noted by the Reviewer, along all the figure legends we repeatedly state that data are mean +/- SEM, which was not right/accurate in all the cases. We appreciate and agree with the Reviewer's comment and we apologize for this oversight. We have revised all the figure legends and now provide the correct description of how data have been analyzed.

3- It would be clearer if all flow cytometry data expressed mean intensity fluorescence (MIF) or % populations not both. Furthermore, in relation to the flow cytometry data please note the following:

- a. Gating schemes should always be shown and if not shown in text needs to be shown in supplemental data**
- b. Methods used to gate - based on unstained samples, fluorescence minus one controls, isotype controls etc.**
- c. How compensation was carried out? Multi-color flow cytometry usually uses stained beads**
- d. If using MFI - compare means/medians, its important to show that the flow cytometer machine has been calibrated on a regular basis otherwise MFIs can change over time.**
- e. Example of flow data in graphs presented shown in plots in supplemental data, if not in body of text.**

We apologize for the confusion that we have generated to the Reviewer when analyzing flow cytometry data. We use % of population to provide quantitative results in terms of frequency (*i.e.*, percentage), and we use geometric mean intensity of fluorescence (M.I.F.) to determine the grade of positivity (*i.e.*, fluorescence intensity). For the latter the expression data for cell populations of interest are presented as a relative M.F.I., which is the ratio between the M.F.I. value of the sample stained with all the experimental markers and the M.F.I. value of the negative control sample, rather than the percentage of positive events. This kind of analytical approach has the advantage, unlike the percentage, of providing “relative” quantitative information on antigen expression on either the cell surface or in intracellular compartments.

As indicated by the Reviewer we have included a more detailed explanation of the flow cytometry analysis performed in material and method sections. Examples of gating schemes have been included in the main text. Similarly, we have also included example of histograms obtained when analyzing fluorescence intensity.

4- Please include protein sizes in all Western blots.

As indicated, protein sizes in all Western blots have been included.

5- In figure 1C an appropriate control must be included, for example, *Ldlr*^{-/-}*miR*-21^{-/-} with no bone marrow transplant. The control shown in the supplementary data for this is from a *Ldlr*^{-/-}*Cav*-1^{-/-} mouse which is not ideal.

We appreciate the attention given to details by the Reviewer. The controls used, as wisely noted by the reviewer, were *Ldlr*^{-/-}*miR*-21^{-/-} mice. This was a typo and has been accordingly corrected. This control is now included in Fig 1D.

6- Figure 5- there is no explanation for MERTK hi and lo. Please include a description. More importantly, the Western blot shown in Figure 5 A does not support the conclusion that total MERTK levels were significantly reduced in MiR21 deficient macrophages. In fact the figure shown in Fig 5A to me looks like there is no change. Please replace with a blot that clearly shows the decrease in expression. Same comment for Figure 5F- the increase in iNOS is not evident in the blot shown.

We apologize for the confusion we created due to the lack of description of “hi” and “low”. We were referring to high exposure and low exposure, respectively. Since the quality of the selected blots was sub optimal, we have, as indicated by the Reviewer, replaced this blot. The new blot can be found in the new Fig 6B. The new blot shows the decrease in the expression in *miR*-21^{-/-} peritoneal macrophages and there is no need to show different exposures to support our conclusions. Similarly, the increase in iNOS is shown in the new Fig 4E.

7- Housekeeping genes used for qRT-PCR are not always stated.

The description the analysis of miRNAs (*miR*-21) by qRT-PCR was missing in the original submission as such the housekeeping control (U6) for this purpose was not included. In the revised version both have been accordingly described and stated in Materials and methods section and corresponding figure legends.

Minor points:

1. *miR-21 is a key signal mediating the balance of pro- and anti-inflammatory responses this should be stated early on, not towards the end of the introduction after discussing it in both cases.*

We agree with the Reviewer and as suggested we have modified introduction and discussion accordingly.

2. *qRT-PCR changes from fold-change to relative expression - please ensure consistency throughout*

This has been revised and modified accordingly. As such we express qRT-PCR data as Relative expression levels normalized to housekeeping control 18 S rRNA for mRNAs and U6 for miRNAs.

3. *p-values not shown on each graph and not stated in figure legends. Only general $p < 0.05$ considered significant mentioned.*

We have included the P values for each graph.

4. *Some graphs show p-value, stars and ns - please ensure consistency in the way the data is presented.*

We have revised every figure legend to provide consistency throughout the manuscript.

5. *Figure 7 should have a figure legend describing the image - every figure should be stand alone.*

As indicated by the Reviewer we have include a description for new Figure 8 (Figure 7 of the initial submission).

6. *Italics for in vivo, in vitro etc.*

This has been corrected.

7. *Please use the correct nomenclature of cytokines i.e TNF- α , IL-6, IL-1 β , NF- κ B etc. not TNF α , IL6, IL1 β , NF-KB*

Nomenclature has been corrected

8. *Please correct typographical errors for example Sheedy et. al. not Sheddy et. al.*

Typographical errors including the one exemplified have been corrected

Referee #3:

We would like to thank the Reviewer for writing that our “findings add to the current understanding of both atherosclerotic progression and miR-21 function.” We are grateful for the comments and suggestions received, since they have undoubtedly improved our manuscript. Your comments are italicized and followed by our answers.

1.) *Figure 1c shows miR-21 in situ hybridization (green staining) in plaques of Ldlr-/-miR-21-/- mice transplanted with WT bone marrow but Fig. S1 also shows significant green staining in what are described as Ldlr-/-miR-21-/- mice that were not transplanted. While the plaque area in Fig S1 may be devoid of staining there is significant staining elsewhere in the image despite the fact that the mice lack miR-21. I assume the mice are miR-21 null and not a conditional strain as that is all that appears on Jackson Labs website at this time but I did not see that actually noted in the manuscript. This should be clarified.*

As noted by the Reviewer our *in situ* hybridization in negative control had a substantial amount of unspecific staining. We have improved our *in situ* hybridization technique and the negative control is as expected devoid of staining elsewhere as expected for miR-21 null mice. As indicated we have included the description of the mouse strain used in Materials and Methods section.

2.) The figure legend for 1d makes no mention of the included Oil Red O images or the associated quantification.

We apologize for the missing information that has now been included in the revised version of the manuscript.

3.) The lesion area in the cross sectional images in Fig. 1d is greater in the miR-21-/- transplanted mice although there is apparently no significant difference in the oil red o staining within the cross sectional lesion area. However, in Figure 1 E the 3 images labeled as WT->Ldlr-/- appear to have more oil red o staining in the aortic and thoracic arch regions while the associated quantification shows the opposite i.e. less oil red o in the WT transplanted mice. I expect the labels in 1 E may be switched.

A couple of days after the initial submission of the manuscript we realized that the labels were switched as the Reviewer noticed. We sent to the editorial office the correct Figure 1. We assume that the Reviewer did not receive it and we apologized for the confusion this might have created.

4.) Figure 2A. Was the zoomed in section (right panel) or the whole image (left panel) used to quantify the necrotic core and fibrous cap regions? If the zoomed in section, why was this specific region of the plaque chosen to quantify?

We apologize for not including a description of the criteria to quantify necrotic core and fibrous cap. In the new version of the manuscript we have described within Materials and Methods section a detailed description for the quantification of necrotic core and fibrous cap.

Briefly: The fibrous cap and necrotic core area were measured as a percentage of the total plaque area from the 3 sections from the same mouse. Necrotic core was defined as a clear area that was H&E free (Seimon, Wang et al., 2009). Boundary lines were drawn around these regions, and the area measurements were obtained by image analysis software. Fibrous cap thickness was quantified by choosing the largest necrotic core from triplicate sections and taking a measurement from the thinnest part of the cap, determined by measuring the area between the outer edge of the cap and the necrotic core boundary (Seimon et al., 2009).

5.) The figure legend for Fig 2 states "all data represent mean+/-SEM (n=9-11 per group)" yet the TUNEL+ graph in 2b shows on 7 WT and 6 mir-21 data points.

We appreciate the attention given to detail by the Reviewer. The figure legend has been corrected with accurate number of animals.

6.) The y axis in 3A has a break between 4 and 5. Why? I assume the Y axis of the plots in 1C should be "Lineage PE" (not lineage PE) although I didn't see that that antibody cocktail mentioned in the methods.

The reviewer is right. There is no need for a break in the Y axis of Figure 3A and this has been removed. The typo in the Y axis of figure 1C has been corrected as well and the antibody cocktail has been included in the materials and methods section.

7.) The sheer number of bands in figure 4 makes it difficult to identify the important differences between conditions/ strains. I do not see value in showing the primary western data for so many proteins that do not appear to change across the conditions. There is also no quantification of the differences shown or referenced in the text.

We agree with the Reviewer and we have included the western blots of the protein that do not appear to change in a supplement. The proteins that change remain in figure 4 and we have included the pertinent quantification, as indicated by the Reviewer.

8.) No quantification of the MERTK levels shown in 5A is given but the text describes it as significantly reduced in miR-21 KO cells. The images in my copy do not look significantly different.

This was a concern also raised by Reviewer #2 (Major point 5). As indicated we have replaced the low quality blot.

9.) 5b - What does the # in the panel represent? It is not referenced in the figure legend.

We apologize for the poor description of the comparisons in figure 5 b. We have now included a better description of the comparisons carried out for statistical purposes.

10.) 5C why is COX2 shown here? How does that data compare to the COX2 data shown in 5F with respect to differences observed between WT and KO cells?

We regret that we missed to describe the reason why we included COX2 in Fig 5C, as noted by the Reviewer. COX2 was used as positive control of LPS treatment. We have included this description in the new Fig 6 B and D. The confusion about how the data from Fig. 5C related to 5F came from the fact that we described the effect on inflammation later on, as noted by Reviewer #2 (minor point 1). In the new version of the manuscript we described the effect on inflammation in Fig 4 D and E, thus providing a better flow to the manuscript.

11.) The differences between WT and miR-21-/- in Figure 6 appear to be relatively small across all the panels but the nearly 2x difference in ABCG1 levels at 24 hrs shown in 6E is particularly hard to appreciate. I think the authors should critically evaluate the necessity and/or alternative presentation of this data.

The quantification of the results shown in Fig 6 E (now Fig 7E) was incorrectly normalized. We apologize for this and have provided the accurate normalization and quantification of the results.

REFERENCES

Albarran-Juarez J, Kaur H, Grimm M, Offermanns S, Wettschureck N (2016) Lineage tracing of cells involved in atherosclerosis. *Atherosclerosis* 251: 445-53.

Clausen BE, Burkhardt C, Reith W, Renkawitz R, Forster I (1999) Conditional gene targeting in macrophages and granulocytes using LysMcre mice. *Transgenic Res* 8: 265-77.

Cross M, Mangelsdorf I, Wedel A, Renkawitz R (1988) Mouse lysozyme M gene: isolation, characterization, and expression studies. *Proc Natl Acad Sci U S A* 85: 6232-6.

Seimon TA, Wang Y, Han S, Senokuchi T, Schrijvers DM, Kuriakose G, Tall AR, Tabas IA (2009) Macrophage deficiency of p38alpha MAPK promotes apoptosis and plaque necrosis in advanced atherosclerotic lesions in mice. *J Clin Invest* 119: 886-98.

Weber C, Zernecke A, Libby P (2008) The multifaceted contributions of leukocyte subsets to atherosclerosis: lessons from mouse models. *Nat Rev Immunol* 8: 802-15

2nd Editorial Decision

17 May 2017

Thank you for your patience. We have now received the enclosed reports from the referees that were asked to re-assess it. As you will see the reviewers are now globally supportive and I am pleased to inform you that we will be able to accept your manuscript pending the following final amendments:

- 1) Please address the minor issues commented by referee 3. Please provide a letter INCLUDING the reviewer's reports and your detailed responses to their comments (as Word file).
- 2) Please carefully check the authors guidelines for formatting your supplemental information: In your case, the pdf file should be relabeled Appendix (see:

<http://embomolmed.embopress.org/authorguide#expandedview>), including within the file for every individual items; a Table of content should be included on the 1st page. Please refer to each item as Appendix Figure S1 or Appendix Table S1 and so on in the main article.

3) please include the RNAseq accession number within the main article.

Please submit your revised manuscript within two weeks. I look forward to seeing a revised form of your manuscript as soon as possible.

***** Reviewer's comments *****

Referee #1 (Remarks):

My comments were addressed adequately.

Referee #2 (Comments on Novelty/Model System):

The authors have employed the most relevant models available to define a role for hematopoietic miR-21 in atherosclerosis and specifically in regulating macrophage apoptosis, and efferocytosis. This work was elegantly performed using BMT and the justification for this has been well described in the response to reviewers comments.

Referee #2 (Remarks):

The authors have carefully addressed all my initial comments and concerns. Specifically they have now included appropriate statistical analysis namely an unpaired two sided Student's t-test, and/or one-way ANOVA with Bonferroni correction for multiple comparisons and Mann-Whitney test where relevant.

They have also included more detailed explanation of the flow cytometry analysis performed in material and method sections. Furthermore, examples of gating schemes have been included in the main text as requested. They have revised both the introduction and discussion to explain the role of miR21 in mediating the balance of pro- and anti-inflammatory responses. All minor comments have also been addressed.

Referee #3 (Remarks):

The authors have addressed the majority of issues that were raised during the initial review. However, I think the following issues still need to be addressed.

- 1.) The manuscript would benefit from a careful proofread as there are numerous instances of missing words or typos.
- 2.) The methods section says that the aortic lesion size was obtained by averaging the lesion areas in 4 sections while only 3 sections (per methods and figure legend) were used to determine fibrous cap and necrotic core area. Necrotic core area is represented as % of lesion area so shouldn't the same number of sections be used for calculating each area?
- 3.) Why, within a given experimental condition, are different numbers of data points shown in the graphs in figure 3a? For example, WT>Ldlr^{-/-} includes 9 points for necrotic area and 11 for fibrous cap. Are there instances where you measure fibrous cap area of a lesion but don't determine the necrotic area of that same lesion?
- 4.) Why are 14 data points shown in Figure 3a for mir21^{-/-}>Ldlr^{-/-} fibrous cap area when the figure legend says 9-11 animals per group? How many animals in total were transplanted with wt and mir21^{-/-} bone marrow, respectively? Was only a subset of each group used for the analysis shown in any given figure? If yes, how was it determined which animals would be included in each analysis.
- 5.) I think the label on the left panels in figure 1d should be WT->Ldlr^{-/-} and the y axis labels in figure 6c and 7c should read "MFI" not "MIF".

We are grateful to the three Reviewers for supporting the publication of our work.

Specific responses to Reviewer #3 are outlined. Comments are italicized followed by our answers:

The authors have addressed the majority of issues that were raised during the initial review. However, I think the following issues still need to be addressed.

We would like to thank the Reviewer for finding that we addressed the majority of the issues found in the initial review. We are very appreciative for the careful second revision, which has undoubtedly further strengthen our manuscript.

- 1.) *The manuscript would benefit from a careful proofread, as there are numerous instances of missing words or typos.*

We apologize for the missing word and types. The manuscript has been carefully proofread and edited accordingly.

- 2.) *The methods section says that the aortic lesion size was obtained by averaging the lesion areas in 4 sections while only 3 sections (per methods and figure legend) were used to determine fibrous cap and necrotic core area. Necrotic core area is represented as % of lesion area so shouldn't the same number of sections be used for calculating each area?*

We apologized for the mistake. 3 sections were used for the quantifications. This has been corrected within the manuscript.

- 3.) *Why, within a given experimental condition, are different numbers of data points shown in the graphs in figure 3a? For example, WT>Ldlr-/- includes 9 points for necrotic area and 11 for fibrous cap. Are there instances where you measure fibrous cap area of a lesion but don't determine the necrotic area of that same lesion?*

We thank the Reviewer the attention given to the details. Same numbers of animals were used to determine necrotic core and fibrous cap and are now included in the new Figure along with correct *p* value.

- 4.) *Why are 14 data points shown in Figure 3a for mir21-/->Ldlr-/- fibrous cap area when the figure legend says 9-11 animals per group? How many animals in total were transplanted with wt and mir21-/- bone marrow, respectively? Was only a subset of each group used for the analysis shown in any given figure? If yes, how was it determined which animals would be included in each analysis.*

The figure legend should have state 11-14. 11 animals were transplanted with *wt* BM and 14 with *mir-21-/-* BM and all animals (see response to point 3) are now included in the analysis and the pertinent *p* value.

- 5.) *I think the label on the left panels in figure 1d should be WT->Ldlr-/- and the y axis labels in figure 6c and 7c should read "MFI" not "MIF".*

The typo has been corrected in the labels of both figures.

Corresponding Author Name: Carlos Fernandez-Hernando and Yajaira Suarez
Journal Submitted to: EMBO Molecular Medicine
Manuscript Number: EMM-2016-07492-V2